# An atypical DYRK kinase connects quorum-sensing with posttranscriptional gene regulation in *Trypanosoma brucei*

**Mathieu Cayla, Lindsay McDonald, Paula MacGregor[†], Keith Matthews***

Institute for Immunology and Infection Research, School of Biological Sciences, Charlotte Auerbach Road, University of Edinburgh, Edinburgh, United Kingdom

**Abstract** The sleeping sickness parasite, *Trypanosoma brucei*, uses quorum sensing (QS) to balance proliferation and transmission potential in the mammal bloodstream. A signal transduction cascade regulates this process, a component of which is a divergent member of the DYRK family of protein kinases, TbDYRK. Phylogenetic and mutational analysis in combination with activity and phenotypic assays revealed that TbDYRK exhibits a pre-activated conformation and an atypical HxY activation loop motif, unlike DYRK kinases in other eukaryotes. Phosphoproteomic comparison of TbDYRK null mutants with wild-type parasites identified molecules that operate on both the inhibitory 'slender retainer' and activatory 'stumpy inducer' arms of the QS control pathway. One of these molecules, the RNA-regulator TbZC3H20, regulates parasite QS, this being dependent on the integrity of its TbDYRK phosphorylation site. This analysis reveals fundamental differences to conventional DYRK family regulation and links trypanosome environmental sensing, signal transduction and developmental gene expression in a coherent pathway.

***For correspondence:**
keith.matthews@ed.ac.uk

**Present address:** [†]Department of Biochemistry, University of Cambridge, Cambridge, United Kingdom

**Competing interests:** The authors declare that no competing interests exist.

## Introduction

Eukaryotic cells respond to their environment via signal transduction cascades whose general structure and features have been intensively studied in model eukaryotes. These enable the adaptation to a changing environment or response to external signals that regulate cellular differentiation and specialisation. Many of these signalling cascades have been dissected in detail, revealing broadly conserved networks that regulate molecular function and interactions in evolutionarily separated eukaryotic groups (*Manning et al., 2002a*). Among the most common regulatory mechanisms is through the action of proteins kinases and phosphatases, whose activity controls the reversible phosphorylation of approximately 30% of the proteome of eukaryotes, involving proteins implicated in a wide range of functions, including cell growth and cell-cycle control, cytoskeleton organisation, extracellular signal transmission and cell differentiation. The structure of conventional signalling cascades, however, is not necessarily representative of the diversity of eukaryotic life, with the environmental signalling pathways and regulatory pathways outside commonly studied model eukaryotes being poorly understood.

A tractable model to explore the diversity of eukaryotic signalling networks are kinetoplastid parasites, which separated from the eukaryotic lineage at least 500 million years ago (*Keeling and Burki, 2019*). These organisms encode around 190 protein kinases (*Jones et al., 2014*) and are exquisitely sensitive to their environment. This sensitivity to well-defined environmental stimuli allows the molecular dissection of signalling pathways that drive the differentiation events that characterise the complex life cycles of these organisms. An exemplar is *Trypanosoma brucei*, a kinetoplastid parasite responsible for Human African Trypanosomiasis (HAT) and Animal African Trypanosomiasis (AAT), that is transmitted through the bite of the tsetse fly. One major environmentally signalled event for *T. brucei* involves their differentiation in the mammal host from a replicative 'slender form'

to an arrested and transmission-adapted 'stumpy form' (*MacGregor et al., 2012*). This differentiation is triggered by a quorum-sensing (QS)-like mechanism where parasites respond to the accumulation of a stumpy-induction factor (*Reuner et al., 1997*; *Vassella et al., 1997*; *Rojas et al., 2019*). Once the signal is received, it is transduced via a non-linear hierarchical signalling pathway (*McDonald et al., 2018*) comprising at least 30 molecules (*Mony et al., 2014*). This includes signal processing molecules, protein kinases and phosphatases and posttranscriptional gene expression regulators as well as additional proteins of unknown function.

One of the components involved in the differentiation process is a molecule related to the protein kinase Yak sub family (*McDonald et al., 2018*; *Mony et al., 2014*). Yak kinases belong to the dual-specificity yak-related kinases (DYRK) family included in the CMGC group, which is over represented in trypanosomatids compared to humans (*Parsons et al., 2005*). The DYRK family is subdivided into five sub families, the homeodomain-interacting protein kinases (HIPKs), the pre-mRNA processing protein four kinases (PRP4s), the Yak kinases (present in lower eukaryotes only), the DYRK1 and the DYRK2 kinases (reviewed by *Aranda et al., 2011*). In mammals, the DYRK1/2 sub families are characterised by the DYRK-homology (DH)-box upstream of a kinase core that contains the ATP-binding domain and the activation loop. The activity of DYRK kinases is dependent on auto-phosphorylation of the second tyrosine residue in the YxY motif present in the activation loop, and they phosphorylate substrates on serine and threonine residues. The full activity of mature DYRK proteins may also require other phosphorylation events outside the kinase core (*Cheng et al., 2009*) or may depend on their interaction with other proteins (*Kim et al., 2004*; *Cheng et al., 2009*). Several mammal DYRKs have also been implicated in cell differentiation. For example, human DYRK1A/murine DYRK1B and human DYRK2 have been suggested to have a role in differentiation mediated by cell cycle arrest in G1/G0 and G2/M, respectively (*Maddika and Chen, 2009*; *Yabut et al., 2010*; *Zou et al., 2004*).

Here, we describe the essential role of several unconventional features in the activity of an atypical DYRK-family kinase identified in the parasite *T. brucei* and that plays a major role in the quorum-sensing stimulated development from slender to stumpy forms in the mammalian bloodstream. We also describe the identification of likely *T. brucei* DYRK (TbDYRK) substrates and reveal their involvement in both inhibitory and stimulatory arms of the developmental control pathway. This places TbDYRK at a pivotal position in the control of parasite developmental competence, connecting signal transduction to gene expression regulatory processes.

## Results

### Tb927.10.15020 encodes for a divergent kinase of the DYRK family

To initially analyse the gene Tb927.10.15020, we performed a phylogenetic analysis using the kinase core of all members of the CMGC kinase family from human, *C. elegans*, *D. melanogaster*, *S. cerevisiae* and all identified members of the DYRK family from *T. brucei* (*Jones et al., 2014*). This identified a DYRK1 family kinase (Tb927.5.1650) and a previously characterised DYRK2 member (TbDYRK2; Tb927.11.3140; [*Han et al., 2012*]). The analysis revealed that Tb927.10.15020 is a divergent DYRK belonging to a paraphyletic group of DYRK2 (*Figure 1a*). Indeed, the two *C. elegans* kinases with which the trypanosome protein clusters, Ce_C36B7.1 and Ce_C36B7.2, have been identified as DYRK2 kinases in the kinase.com database (http://kinase.com) and as DYRK3 in the wormbase.org database (https://wormbase.org) (*Manning, 2005*). It is also notable that Tb927.10.15020 presents a potential divergent DH box (NEx$_{(2)}$DDx$_{(3)}$Y) (*Becker et al., 1998*) specific for DYRK1/2, a divergent NAPA-1 region (Lx$_{(3)}$Ex$_{(2)}$Ex$_{(15)}$G), but no NAPA-2, both specific for DYRK2 and present in the previously analysed TbDYRK2 (*Han et al., 2012*; *Kinstrie et al., 2010*; *Figure 1b*). Multiple sequence alignment of the kinase core of Tb927.10.15020 with several well characterised members of the DYRK family from a number of model organisms (*Figure 1—figure supplement 1*), also revealed the presence of three long inserts in the trypanosome kinase. The kinase is present in the genome of different trypanosomatid organisms and sequences of the first two inserts are relatively well conserved among these orthologues, while the sequence of the last insert in Tb927.10.15020 is poorly conserved outside the *brucei* group (*Figure 1—figure supplement 2*). These multiple sequence alignments also revealed the presence of a serine (S856) instead of the classical glycine in the highly conserved DFG motif (*Figure 1—figure supplement 1*), as well

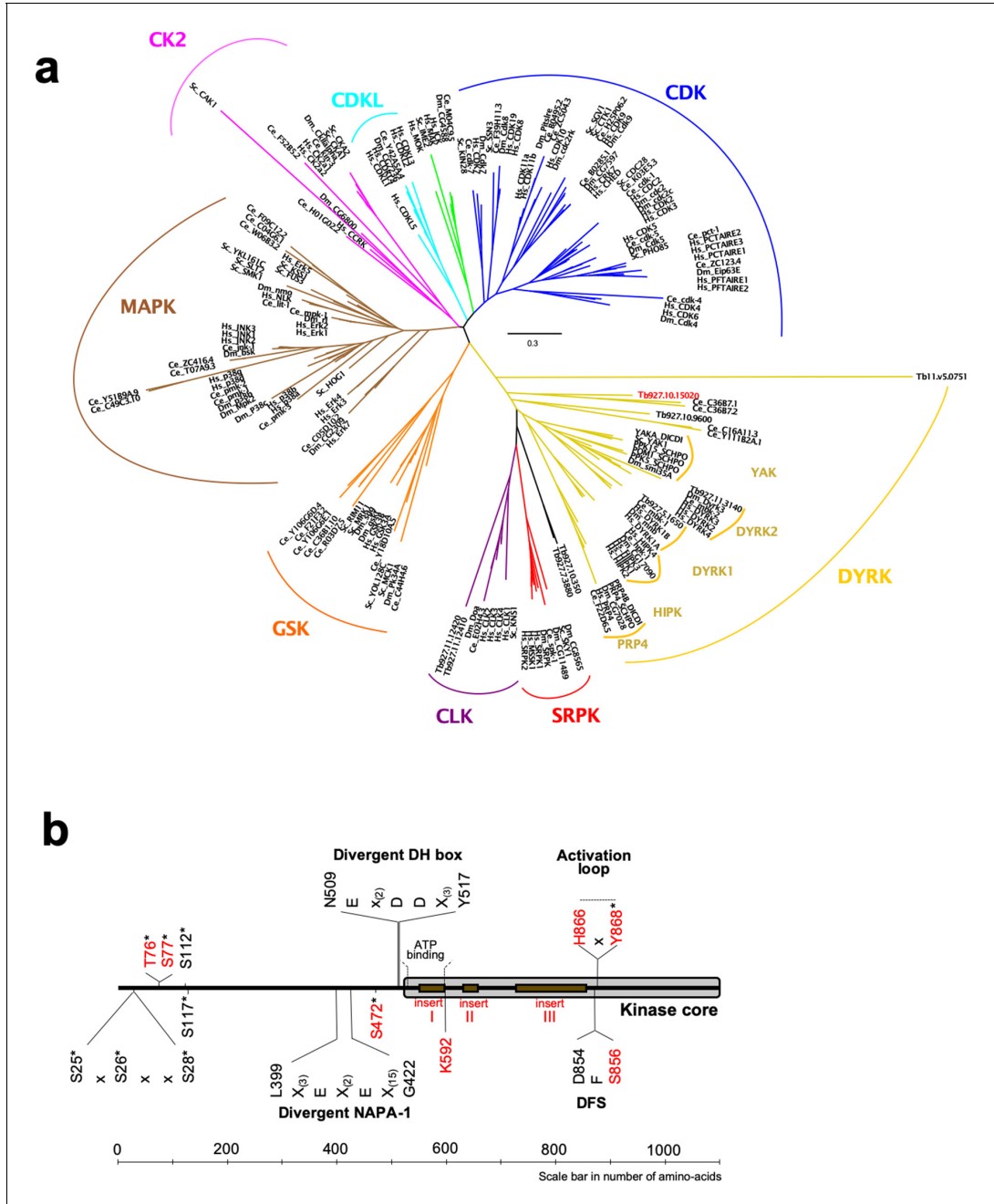

**Figure 1.** Phylogenetic analysis of the CMGC protein kinase family. (a) The evolutionary history was inferred by using the Maximum Likelihood method based on the Whelan And Goldman + Freq. model (*Whelan and Goldman, 2001*). Initial tree(s) for the heuristic search were obtained by applying the Neighbour-Joining method to a matrix of pairwise distances estimated using a JTT model. The tree is drawn to scale, with branch lengths measured in the number of substitutions per site. All positions with less than 95% site coverage were eliminated. That is, fewer than 5% alignment gaps, missing data, and ambiguous bases were allowed at any position. Hs = *Homo sapiens*, Ce = *Caenorhabditis elegans*, Dm = *Drosophila melanogaster*, Sc = *Saccharomyces cerevisiae*, DICDI = *Dictyostelium discoideum*, Tb927 = *Trypanosoma brucei*. The early divergent TbDYRK is highlighted in red. (b) Schematic representation of linear protein sequence of TbDYRK, highlighting particular characteristics of its sequence. Identified phosphosites are represented by an *. Insertions I, II and III are presented with the brown boxes. All residues or inserts mutated or deleted in this study are represented in red.

The online version of this article includes the following figure supplement(s) for figure 1:

**Figure supplement 1.** Multiple sequence alignment of the kinase core of TbDYRK (lower sequence) with other DYRKs from other species.

**Figure supplement 2.** Phylogenetic analysis and conservation of DYRK orthologues in other kinetoplastids.

as the presence of a histidine (H866), where usually is found a phosphorylable residue, in the HxY motif of the activation loop (*Figure 1—figure supplement 1*). Analysis of this unconventional activation loop in various species revealed the motif was absent in yeast DYRKs, whereas one of 11 human DYRKs present such a motif and one of six in *Drosophila.* In contrast, of the seven DYRKs identified in the *T. brucei* kinome (*Jones et al., 2014*), four present unconventional activation motifs, of which two include a histidine (Tb927.10.15020 and Tb927.5.1650), suggesting a particular mode of regulation in trypanosomes more developed than in other species. Given the association with the DYRK family, we henceforth refer to the protein encoded by Tb927.10.15020 as TbDYRK (previously termed 'YAK' in *Mony et al., 2014* and *McDonald et al., 2018*).

## TbDYRK activity is required for the slender to stumpy differentiation

We initially confirmed the role for TbDYRK in differentiation competence described by *Mony et al. (2014)* and *McDonald et al. (2018)*. A wild-type differentiation competent cell line and a previously validated TbDYRK knock-out (KO) cell line (*McDonald et al., 2018*) were assayed in vitro using the cell permeable cyclic AMP analogue, 8-pCPT-cAMP, that promotes the expression of some characteristics of stumpy forms (*Laxman et al., 2006*; *MacGregor and Matthews, 2012*). As expected, the wild-type cells differentiated in response to treatment, causing a 60% growth inhibition at 96 hr compared to untreated cells, whereas the TbDYRK KO cells had reduced differentiation competence, with only 23% growth inhibition at the same timepoint (*Figure 2a*). To gain further insight into the specific function of this kinase and the role of selected residues for its kinase activity, we also created two further cell lines ectopically expressing either a TY-YFP-tagged non-mutated version of the kinase ('NM') or a predicted 'kinase dead' mutant with lysine 592 converted to alanine ('K592A'); in each case these were expressed in cells retaining the endogenous gene. Importantly, the ectopic copy of the gene was provided with an aldolase 3'UTR and can therefore present an expression and regulatory profile different to the endogenous gene. We then induced ectopic kinase expression using doxycycline and compared the response of each cell line to 8-pCPT-cAMP. Ectopic expression of the NM version of the kinase resulted in an increased differentiation response to 8-pCPT-cAMP compared to wild-type cells (86% growth inhibition at 96 hr), whereas ectopic expression of the K592A mutant considerably reduced differentiation (27% growth inhibition at 96 hr), despite the presence of the endogenous wild-type gene, presumably through a dominant-negative mechanism (*Figure 2a*).

Levels of TbDYRK protein expression could not be detected using an antibody raised against TbDYRK or even using epitope tagging. Therefore, as an alternative to monitor the level of expressed kinase, we measured the total amount of TbDYRK mRNA by RT-qPCR, either detecting both the endogenous and the ectopic gene-derived mRNAs or, by specifically targeting the YFP component, only detecting the ectopic gene-derived mRNA (*Figure 2a*, middle panel). Interestingly, when TbDYRK expression was examined in the NM kinase line, the total amount of TbDYRK transcript did not exceed the level in WT cells (74 ± 18.1% of expression), despite the ectopically expressed gene contributing 31 ± 6.9% of the total TbDYRK mRNA. These results suggest that the ectopic expression of the NM kinase leads to a reduction of the endogenous mRNA and that the total amount of mRNA may be regulated. With the expression of the K592A mutant, the total amount of mRNA was 129 ± 3.1% compared to WT cells, of which 60 ± 6.3% corresponded to the K592A-YFP fusion.

Next, we determined the activity of the NM and predicted kinase-dead mutant in vitro. To identify a suitable substrate for this assay, we purified the expressed tagged NM kinase from insect cells and performed an in vitro kinase assay in the presence of a panel of candidate generic substrates (*Figure 2—figure supplement 1a*). The first 200 amino acids of the *Mus musculus* caspase 9 (Casp9) was the best available substrate, and was used to perform kinetic analysis to determine the optimal conditions to assay TbDYRK activity (*Figure 2—figure supplement 1b and c*). We then applied this methodology to measure the phosphotransferase activity of the NM and K592A mutants, also expressed in and purified from insect cells. The mean of the activity measured with the NM kinase was set at 100% and the results indicated that the mutant K592A exhibited reduced activity compared to NM (48.2 ± 8.0%, *Figure 2a*, lower panel), demonstrating that the predicted 'kinase dead' mutation reduced but did not completely eliminate activity.

To confirm the in vitro results suggesting that the ectopic expression of the active NM kinase leads the cells to arrest at a lower concentration and potentially drives the cells to stumpy-like

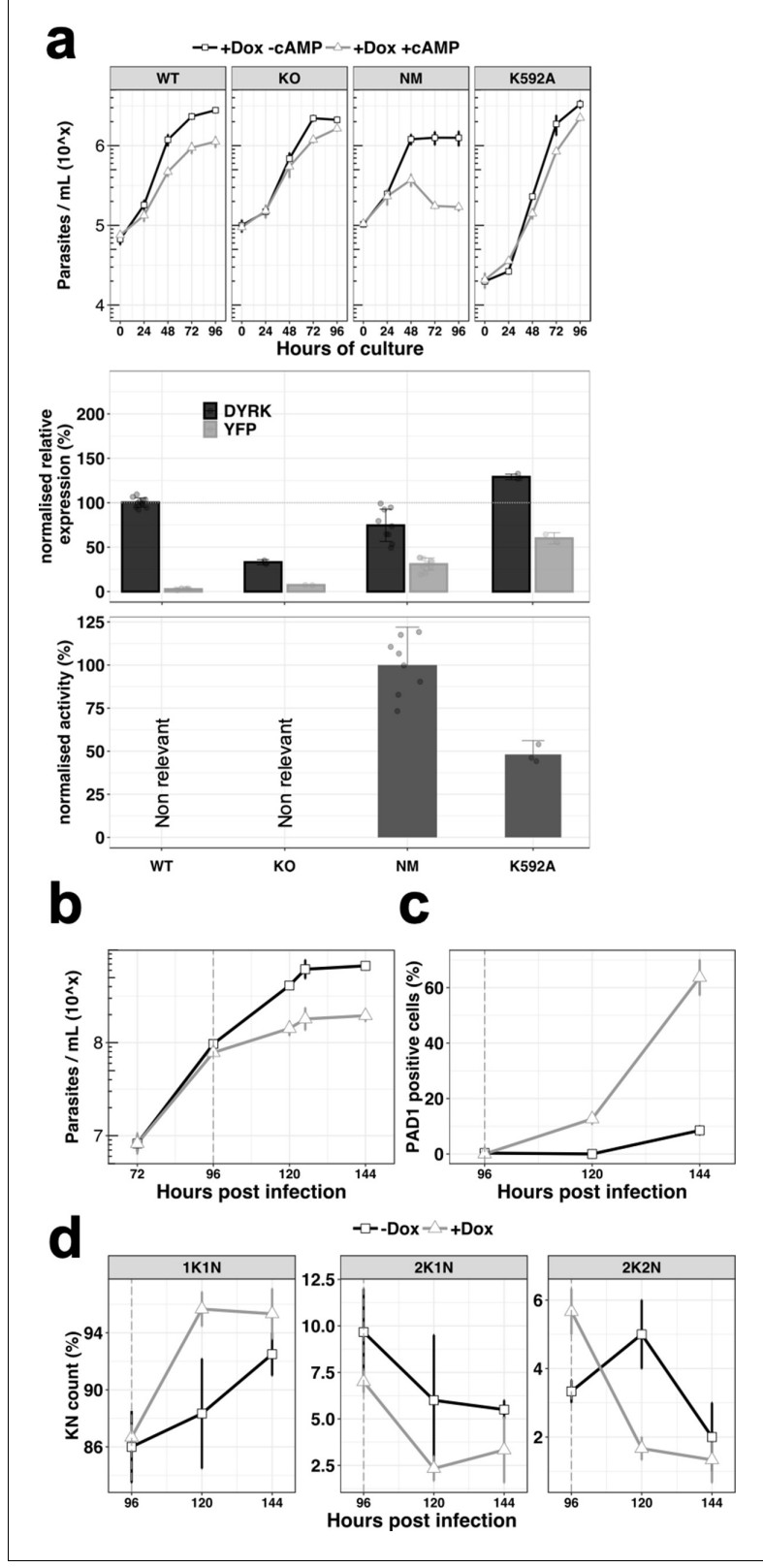

**Figure 2.** Ectopic expression of the TbDYRK drives stumpy differentiation. (**a**) The top panel shows the in vitro phenotype analysis after expression of TbDYRK in different strains treated with doxycycline in the presence (grey) or absence (black) of 8-pCPT-cAMP (n = at least two experiments in three replicates, error bars = SEM). WT = Parental cell line, KO = Knock out cell line for the gene TbDYRK, NM = Ectopic expression of the non-

*Figure 2 continued on next page*

*Figure 2 continued*

mutated version of TbDYRK, K592A = Ectopic expression of TbDYRK carrying the mutation K592A. The middle panel represents the mRNA level of expression of TbDYRK (in black), including the endogenous and the ectopic gene when present, and of the YFP tag of the ectopic fusion gene (in grey) (n = 1 experiment in three replicates; error bars = SEM; individual datapoints are also presented). The dotted horizontal grey line represents 100% of expression (mean), obtained from the expression of TbDYRK in the WT cell line after 24 hr of incubation with doxycycline and no 8-pCPT-cAMP. Note that background signal by RT-qPCR is retained with the TbDYRK KO cell line; independent northern blotting has confirmed absence of the transcript. The lower panel represents the activity of the purified kinase against the generic substrate Casp9 as measured by radioactive kinase assay. The mean level of activity of different experiments of NM has been set at 100% of activity (n = at least two experiments in three replicates; error bars = SEM; individual datapoints are also presented). Statistical p-values are provided in *Supplementary file 5* for the kinase activity and mRNA level expression. (b) Parasitaemia measured in the bloodstream of mice infected (n = 3, error bars = SEM) with the NM strain and treated (+Dox, grey) or not (-Dox, black) after 96 hr post infection. (c) Percentage of PAD1 positive cells in the same blood smears as in panel b. (d) K (inetoplast) N(uclear) scoring from DAPI staining of blood smears of the mice infected as previously described. n = 250 cells, error bars = SEM.

The online version of this article includes the following figure supplement(s) for figure 2:

**Figure supplement 1.** Kinase activity assays for TbDYRK.

differentiation earlier, we infected mice with the NM strain. The data in *Figure 2b* demonstrates that the parasitaemia was reduced after induction of the expression of the non-mutated kinase by provision of doxycycline at 96 hr post-infection. Further, the cells presented a strong increase of PAD1 expression as assessed by IFA (*Figure 2c*) (63.7 ± 11.0% versus 8.5 ± 2.1% at 144 hr post-infection, uninduced and induced, respectively) and arrested in G1/G0 earlier (*Figure 2d*). These results show that the ectopic expression of active TbDYRK drives the cells to differentiate to stumpy forms at a lower parasitaemia.

## N-terminal phosphosite-residues and the HxY motif of the activation loop are essential for TbDYRK activity

We next analysed the role of selected residues and domains of the TbDYRK molecule to explore their difference from conventional eukaryotic DYRKs. First, we and others have identified nine phospho-sites on the TbDYRK protein (asterisks on *Figure 1b*; *McDonald et al., 2018*; *Nett et al., 2009*; *Urbaniak et al., 2013*). Eight of these are residues located in the N-terminal domain of the protein, and one, Y868, belongs to the atypical HxY motif of the activation loop. We assessed the role of three of the N-terminal residues for kinase function, that is T76, S77 and S472, by mutating them to alanine and ectopically expressing the mutants in cells retaining the endogenous gene copies. Thereafter, similar analyses to those previously were used, namely i) scoring cell growth upon ectopic expression of the mutants in the presence or absence of 8-pCPT-cAMP in vitro (*Figure 3a*), ii) quantification of the mRNA level of each mutant and total TbDYRK (*Figure 3b*), and iii) quantification of the phosphotransferase activity of each protein expressed in insect cells (*Figure 3c*). The results obtained for the WT, KO and NM repeat those presented in *Figure 2* to provide comparison with the generated mutants; the remainder are a combination of at least three independent experiments in three replicates for each mutant. Expression of the T76A and S77A mutants did not change the response to 8-pCPT-cAMP of the cells compared to WT parasites although the S77A mutant grew more slowly (*Figure 3a*). Both mutations render the kinase completely inactive (*Figure 3c*) and their mRNAs were well expressed (*Figure 3b*). For both mutations, mRNAs derived from the mutated ectopic copy represented the majority of the total TbDYRK mRNA in each cell line (72 ± 6.3, 72 ± 1.27%), although the combined mRNA level was not elevated over wild-type abundance, implying that expression of the mutants causes regulation of transcripts derived from the endogenous gene as seen earlier. The mutation of S472, by an alanine, also completely disrupted the kinase activity and, similar to the phenotype observed with the KO cell line, generated resistance to 8-pCPT-cAMP treatment, suggesting the inactive expressed kinase prevented the function of TbDYRK expressed from the endogenous locus (*Figure 3*). This phenotype could be contributed to by the high level of expression of the tagged gene, with the combined endogenous and ectopically expressed TbDYRK mRNA reaching 164 ± 11.11% of TbDYRK in wild type cells.

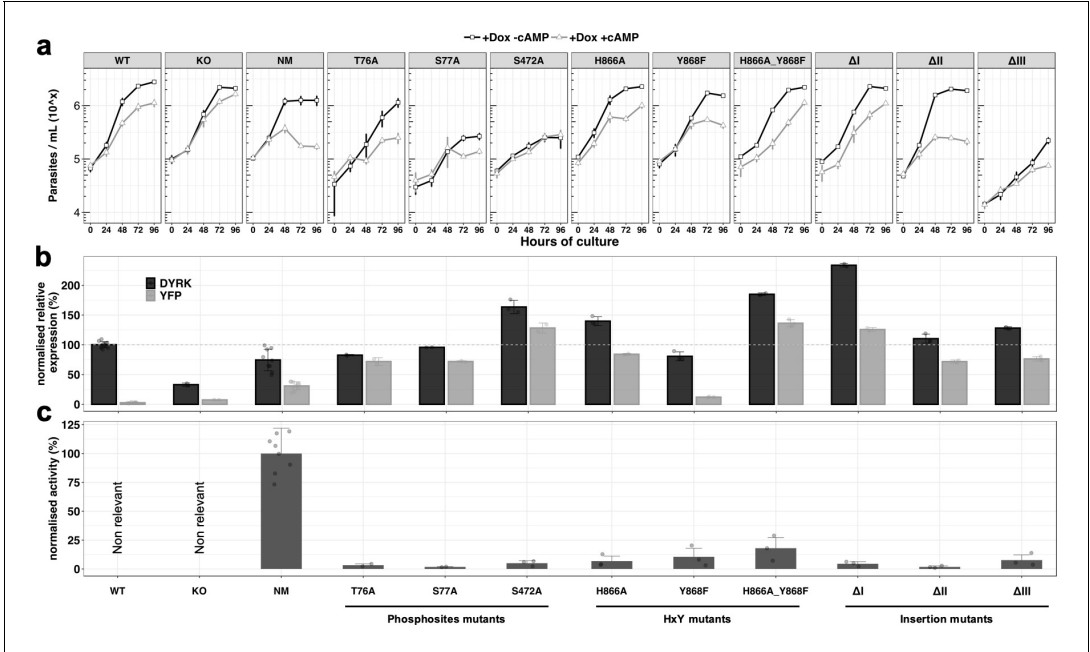

**Figure 3.** Analysis of the unconventional features of TbDYRK with respect to their phenotype, expression and activity. (a) In vitro phenotype analysis after expression of TbDYRK mutants treated with doxycycline in the presence (grey) or absence (black) of 8-pCPT-cAMP (n = at least 2 experiments in three replicates; error bars = SD). WT = Parental cell line, KO = Knock out cell line for the gene TbDYRK, NM = Ectopic expression of the non-mutated version of TbDYRK. WT, KO and NM results are the same as presented in *Figure 2* and are used here as control; other columns reflect the respective mutants analysed. (b) mRNA level of expression of TbDYRK (in black), including the endogenous and the ectopic gene when present, and of the YFP tag of the ectopic fusion gene (in grey) (n = 1 experiment in three replicates, error bars = SEM). The dotted horizontal grey line represents 100% of expression (mean), obtained from the expression of TbDYRK in the WT cell line after 24 hr of incubation with doxycycline and no 8-pCPT-cAMP. Statistical p-values are provided in *Supplementary file 5*. (c) Activity of the purified kinase against the generic substrate Casp9 as measured by radioactive kinase assay. The mean level of activity of different experiments of NM has been set at 100% of activity (n = at least 2 experiments in three replicates; error bars = SEM). Statistical p-values are provided in *Supplementary file 5*.

The online version of this article includes the following figure supplement(s) for figure 3:

**Figure supplement 1.** Modelling of the kinase core of TbDYRK.

**Figure supplement 2.** Analysis of cell viability by alamar blue of parasites expressing the ectopic DYRK NM/H866A / ΔII in response to 8-pCPT-cAMP treatment.

We next investigated the function of the unconventional HxY motif in the activation loop, where a phosphorylable residue (Y or T) is usually observed instead of the histidine residue in TbDYRK. As previously, we analysed the effect of the mutations H866A, Y868F and the combination H866A_Y868F on the response to 8-pCPT-cAMP treatment, on TbDYRK mRNA levels and on kinase activity (*Figure 3a–c*). Mutations of these residues disrupt the kinase activity with an activity level of 6.84 ± 4.34% (H866A), 10.54 ± 7.4% (Y868F) and 18.02 ± 9.25% (H866A_Y868F) compared to the non-mutated (NM) protein. Expression of mutant H866A led to a similar response as WT cells to 8-pCPT-cAMP. The TbDYRK Y868F mutant mRNA was expressed at a lower level than the NM mRNA (12.05 ± 0.73%), and the cells responded to 8-pCPT-cAMP similarly to WT cells. In contrast, expression of a mutant with both mutations (H866A_Y868F) led to a slight resistance to 8-pCPT-cAMP, with the mutant being highly expressed and comprising the majority of TbDYRK mRNA in the cell, with reduction in the contribution of the mRNA from the endogenous TbDYRK allele, suggesting a negative regulation.

Overall, these data show that mutation of any of the four putative phosphosite-residues, either in the N-terminal domain or the HxY motif in the activation loop, disrupt TbDYRK kinase activity. Despite the different mutations having the same apparent effect of creating kinase-dead mutants, a range of phenotypes (increased, decreased or unchanging responsiveness to 8-pCPT-cAMP) were observed when these were ectopically expressed in cells.

## The three inserts inside the kinase core have a major role in the kinase function

As highlighted earlier, the conventional kinase core sequence of TbDYRK is interrupted by three inserts of 36, 22 and 119 amino-acids (*Figure 1—figure supplement 1*), which are predicted to be positioned peripherally to the core enzyme structure and so amenable to functional analysis by deletion (*Figure 3—figure supplement 1a and b*). Each insert was, therefore, deleted and the expressed mutants assayed for their response to 8-pCPT-cAMP (*Figure 3*; ΔI, II or III). Activity assays demonstrated that each of the three inserts was essential for the phosphotransferase activity of the kinase. The ΔI mutant was highly expressed, and phenotypically, this expression rendered the cells relatively resistant to 8-pCPT-cAMP-induced arrest despite the presence of the endogenous TbDYRK allele. The ΔII mutant was expressed at a similar level as NM; however, despite this and the lack of activity of this mutant, a similar phenotype as the expression of NM was observed. Thus, there was a decrease of the parasite density in response to 8-pCPT-cAMP treatment that was greater than seen with WT cells. In contrast, the inactive ΔIII mutant generated a similar phenotype to WT cells in response to the 8-pCPT-cAMP, despite its effective expression although overall growth was slower.

To explore the phenotype of the ΔII mutant, which matched the ectopic expression of the NM kinase and yet lacked kinase activity we performed a viability assay of cells where the NM or the ΔII mutant were expressed in the presence or absence of 8-pCPT-cAMP (*Figure 3—figure supplement 2*). This revealed that expression of the inactive ΔII mutant resulted in cell death after 8-pCPT-cAMP exposure, whereas those expressing the NM kinase were viable. Apparently therefore, expression of the ΔII mutant is toxic to cells exposed to 8-pCPT-cAMP but not those in the absence of this stimulus.

## The S856 in the DFS motif allows a potential pre-activation state

We next investigated the role of S856 that contributes to the unconventional DFS motif of TbDYRK. This motif, usually comprising DFG, functions in the conformational changes upon kinase activation necessary for the correct alignment of the catalytic residues and the binding of ATP. The mobility brought about by the glycine residue allows the switch of the phenylalanine from the inactive state 'DFG-out' to the active state 'DFG-in' (*Kornev et al., 2006*).

By analysis of the *T. brucei* kinome, we firstly established that unconventional DFG motifs (either absent or mutated on the phenylalanine and/or the glycine) were present on 18.5% of *T.brucei* ePKs, and that eight kinases possess a change of the glycine to either an aspartic acid, an asparagine or a serine (*Figure 4a*). We note that the DFS motif is present on four kinases in *T. brucei* (Tb927.9.1670, Tb927.1015020, Tb927.10.16160 and Tb927.11.5340), representing 2.5% of *T. brucei* ePKs, while it is present in only 0.4% of the ePKs in humans.

To investigate the role of this serine in the activation state of the kinase, we modelled the kinase core of TbDYRK using the i-TASSER server. The model obtained revealed interesting features, although the overall scores highlighted complexity of the predictions (c-score = −2.73, estimated TM-score = 0.40 ± 0.13, estimated RMSD = 14.5 ± 3.7 Å), undoubtedly influenced by the three inserts generating a more important b-factor due to the consequent low alignment coverage and the absence of predicted secondary structure (*Figure 3—figure supplement 1a*). Based on the structural prediction, the N-terminal lobe appears to be well conserved with the presence of the classical beta sheet β1–5 and the αC helix. The C-terminal lobe is composed of the classical beta sheet β6/7 after the hinge region, the α-helix C/D/E/F/G/H and I, and also contains the MK1 insert specific of the CMGC family and the α-helix 7 and 8 (*Kannan and Neuwald, 2004*; *Kannan et al., 2007*; *Manning et al., 2002b*; *Figure 3—figure supplement 1a,b,d*). From this analysis, we predicted that the presence of this serine in place of glycine leads to a pre-activation state of the kinase with a phenylalanine in position 'DFS-in', and the aspartic acid pointing towards the active site, as observed on the crystal structure of active p38 (*Figure 4b*). In addition, and in support of the hypothesis of a pre-active conformation of TbDYRK, the model suggests a close conformation of the ATP binding pocket with the glutamine 607 at 3.5 Å from the lysine 592, in concordance with the 3.8 Å measured between the corresponding amino acid of the active structure of p38 (*Figure 3—figure supplement 1c*). This observation is also accompanied by fully aligned regulatory and catalytic (R- and C-) spines as observed in active kinases (*Kornev et al., 2006*; *Taylor and Kornev, 2011*; *Horjales et al., 2012*; *Figure 3—figure supplement 1d*).

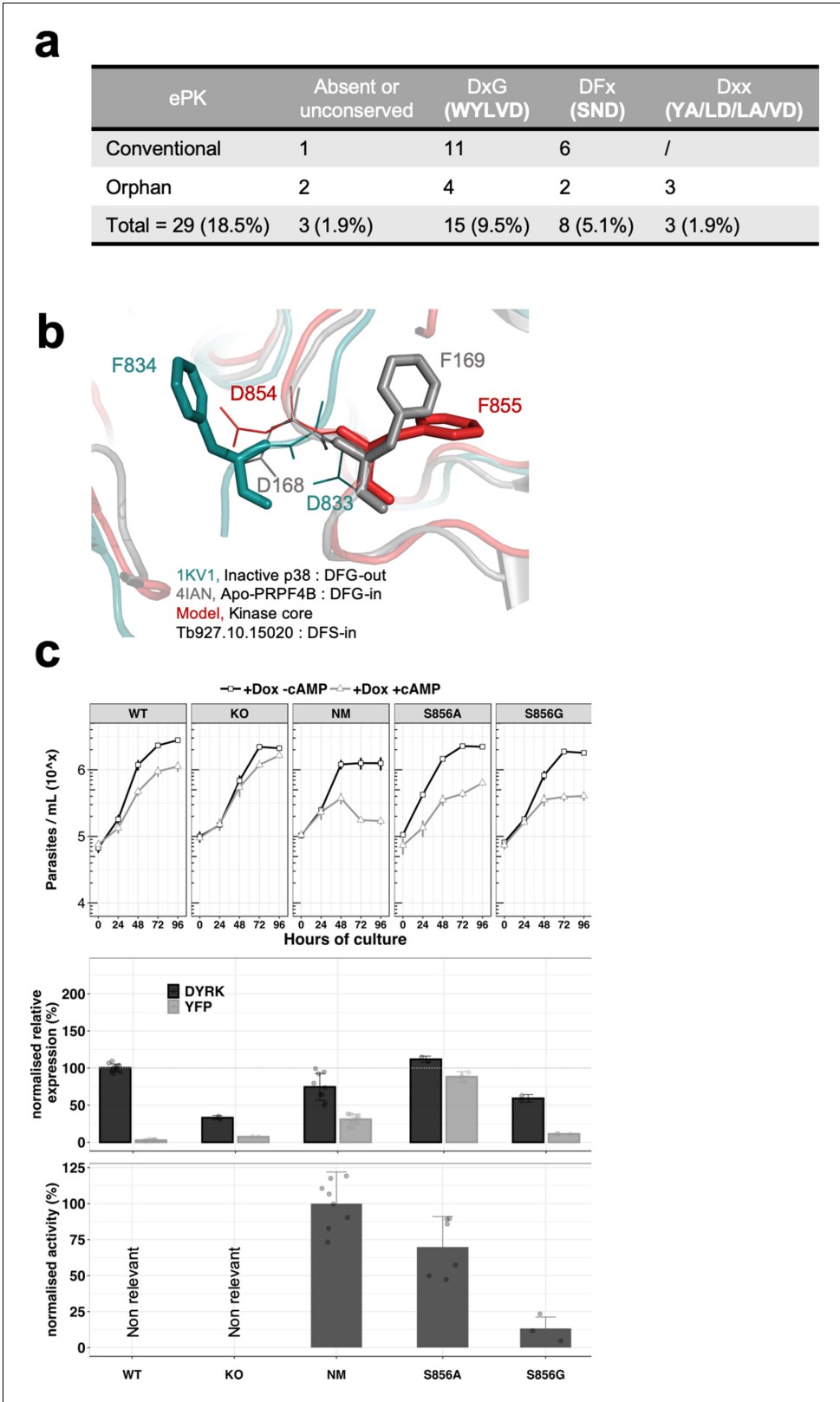

**Figure 4.** The unconventional DSF motif renders the kinase more rigid and is essential for the activity of the kinase. (a) Table indicating the unconventional DFG motif in *T. brucei* ePKs. Conventional = conventional kinases, Orphan = orphan kinases. (b) Model of the TbDYRK kinase core generated by the i-TASSER server, indicating the position 'DFS-in' of the DFS motif of the kinase. (c) Structure/function analysis, as presented earlier, of the DFS

*Figure 4 continued*
motif. WT, KO and NM results are the same as presented in *Figure 2* and are used here as control.
S856A = Ectopic expression of TbDYRK carrying the mutation S856A, S856G = Ectopic expression of TbDYRK
carrying the mutation S856G. For the growth curves, n = 3, error bars = SEM; for the mRNA and kinase activity
assays, n = 3, error bars = SEM. Statistical p-values are provided in *Supplementary file 5* for the kinase activity
and mRNA level expression.

Interestingly, mutation of the serine S856 to alanine (DFA) did not greatly reduce kinase activity (69.83 ± 21.17%, *Figure 4c*, lower panel), whereas the mutation to glycine, to generate a classical DFG motif, almost completely disrupted the kinase activity (13.27 ± 7.96), suggesting that the greater mobility brought by the glycine is detrimental for activity. In addition, despite the strong expression of the S856A mutant (*Figure 4c*, middle panel) and the maintenance of around 70% of its kinase activity in vitro, the same phenotype as WT cell was observed in response to 8-pCPT-cAMP (*Figure 4c*, top panel). These results suggest either that the remaining activity of this mutant is not sufficient to generate the strong NM phenotype, or that the activity against endogenous substrates may be different to activity against Casp9 in vitro.

In combination these structural predictions and functional studies demonstrated that phosphorylation at the N terminus of TbDYRK, the presence of atypical inserts and integrity of the activation loop in the kinase core region, and the presence of the unconventional DFS motif all are important for the activity of the atypical kinase leading to a range of phenotypic outputs in response to 8-pCPT-cAMP. Moreover, there is evidence of mRNA regulation when the mutants were responsive to 8-pCPT-cAMP but less when this response is reduced.

## Identification of substrates of TbDYRK implicated in stumpy differentiation

The next step of the analysis was to explore the signalling pathway in which TbDYRK functions. Therefore, in a first screen, we examined phosphoproteomic changes upon deletion of both TbDYRK alleles in slender form parasites. Parental *T. brucei* EATRO 1125 AnTat 1.1 90:13 and TbDYRK KO cells were cultured in duplicate at equivalent cell density in vitro and protein extracted and analysed after isobaric tandem mass tagging (Flow chart in *Figure 5a*) as described in *McDonald et al. (2018)*. A total of 2499 unique proteins and 7293 unique phosphopeptides were identified; correlations between the replicates were >99% at the peptide level, and at the phosphopeptide level were 0.8805 (*T. brucei* EATRO 1125 AnTat1.1 90:13) and 0.9896 (TbDYRK null mutant) respectively, demonstrating excellent reproducibility (*Figure 5—figure supplement 1*). The results were filtered for peptides with >1.5 fold change in phosphorylation regardless of direction, with an adjusted p value of < 0.05, (*Supplementary file 2*), revealing that 213 peptides on 172 different proteins were less phosphorylated, and that 191 peptides on 149 unique proteins were more phosphorylated in the TbDYRK KO cell line. As expected, the most depleted protein was TbDYRK itself (Tb927.10.15020). Also, supporting the involvement of the TbDYRK protein in the stumpy differentiation pathway, two sites were identified as less phosphorylated in the null mutant line on Protein Associated with Differentiation (PAD) PAD2 (Tb927.7.5940 - S324: Log2–1.276 - adj p=0.0128; S309: Log2–1.268 - adj p=0.0055, *Figure 5a*, *Supplementary file 2*). In addition, the analysis revealed a strong enrichment of proteins implicated in the regulation of gene expression, posttranscriptional regulation, RNA regulation, phosphorylation and protein and vesicle transport. These data indicate that the absence of TbDYRK influences a broad spectrum of substrates implicated in several essential biological processes, particularly those having a direct action on effectors of gene expression such as the eukaryotic translation initiation factor 4e (Tb927.11.11770), the transcription elongation factor s-II, putative (TFIIS2-1, Tb927.2.3580) or the negative regulator of transcription NOT5 protein (Tb927.3.1920, *Figure 5a*), for example.

To filter the extensive list of potential substrates, we took advantage of the structure/function analysis performed previously and used the purified kinase, either as the active (NM) or inactive (H866A) form, to differentially phosphorylate triplicate *T. brucei* procyclic cell lysates previously treated with phosphatase to remove any existing phosphorylation (*Figure 5b*). Lysates of this life cycle stage were used to provide sufficient material for analysis. In this analysis, 38 phophopeptides on 29 unique proteins were significantly phosphorylated (FC > 1.5 – p-value<0.05) with the active

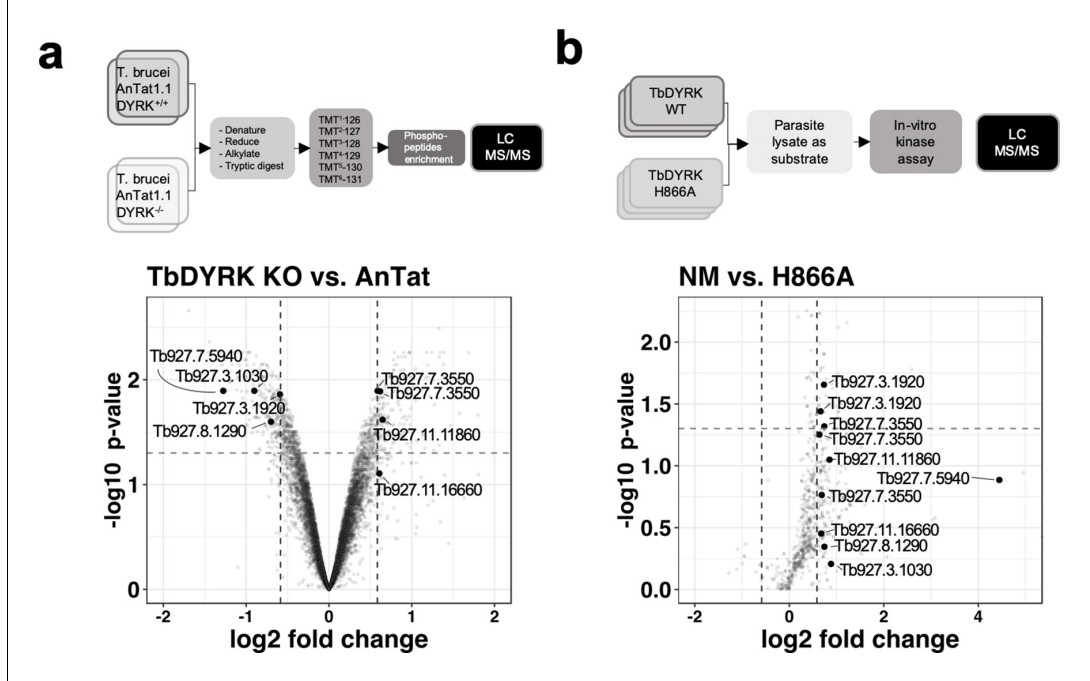

**Figure 5.** Phosphoproteomic analysis for the identification of substrates of TbDYRK. (a) Quantitative phosphoproteomic analysis by TMT isobaric tagging comparing the proteome of WT cells (*T. brucei* AnTat1.1 DYRK+/+) or cells deleted for TbDYRK (DYRK-/-). Top panel: flow chart. Lower panel: Volcano plot of the phosphopeptides with the log2 of the fold change (FC) on the x-axis and the -log10 of the p-value on the y-axis. Vertical dotted lines indicate a |FC| > 1.5 and the horizontal dotted line a p-value<0.05. (b) Quantitative phosphoproteomic analysis comparing the phosphoproteins of cell lysates incubated with the purified active kinase (NM) or inactive (H866A). Top panel: flow chart. Lower panel: Volcano plot of the phosphopeptides with the log2 of the fold change (FC) on the x-axis and the -log10 of the p-value on the y-axis. Vertical dotted lines indicate a |FC| > 1.5 and the horizontal dotted line a p-value<0.05. Common peptides in both analyses, presenting a differential phosphorylation >1.5 times disregarding the sense of regulation, are highlighted in both volcano plots.

The online version of this article includes the following figure supplement(s) for figure 5:

**Figure supplement 1.** Correlation plots between replicates of the phosphoproteomic analysis comparing DYRK-/-cells to WT AnTat1.1 cells.

**Figure supplement 2.** Substrate motif identification.

kinase NM whereas, no significant phosphorylation events were identified with the inactive mutant H866A. As for the previous phosphoproteomic analysis, a GO enrichment of similar functions was also observed in the identified phosphosubstrates that is regulation of gene transcription, transport and cellular reorganisation (*Supplementary file 3*). Comparison of both phosphoproteomic analyses (i.e. whole phosphoproteome and the lysate phosphorylation analysis) demonstrated the enrichment of a kinase substrate consensus motif (R-P/R-x-S/T-P) similar to that of mammalian DYRK 2 and 3 (*Figure 5—figure supplement 2*). Overall, 45 proteins were present in both analyses with a |FC| > 1.5 (*Supplementary file 4*) and eight phosphosites were common for seven proteins (|FC| > 1.5) with two sites being significant with a p value<0.05 (*Figure 5*). These two sites were i) serine 395 of the NOT5 protein, which is less phosphorylated in the KO cell line (Log2FC −0.5910 – p value 0.0138) and phosphorylated by the NM kinase (Log2FC 0.7336 – p value 0.0221), and ii) serine 1048 of the WCB cytoskeleton associated protein Tb927.7.3550, which was phosphorylated in the KO cell line (Log2FC 0.5879 – p value 0.0128) and phosphorylated by the NM kinase (Log2FC 0.7402 – p value 0.0481) (*Figure 5a,b*, *Supplementary file 4*).

We then selected five genes from the 45 proteins present in both proteomic analyses with a |FC| > 1.5 to individually assess their involvement in the TbDYRK-regulated stumpy formation. These were two hypothetical proteins (Tb927.1.4280 implicated in posttranscriptional activation [*Erben et al., 2014*] and Tb927.4.2750), the NOT5 protein previously implicated in the negative regulation of mRNA stability (Tb927.3.1920) (*Erben et al., 2014*; *Lueong et al., 2016*; *Singh et al., 2014*; *Ling et al., 2011*; *Liu et al., 2020*), a zinc finger protein TbZC3H20 (Tb927.7.2660) previously shown to be associated to RBP10 protein, to interact with MKT1 and to be implicated in

the increase of protein translation or in mRNA stability (*Erben et al., 2014*; *Singh et al., 2014*), and a bilobe region protein (Tb927.11.15140). Of the genes analysed, we were only able to generate a null mutant cell line for the TbZC3H20 protein; for all others, conditional RNAi knock-down cell lines were produced. We then investigated the capacity of the cells to differentiate in response to cell permeable 8-pCPT-cAMP or to stumpy induction factor in vivo. Phenotypic analysis of the TbZC3H20 KO cell line in vitro and in vivo demonstrated that this cell line was unable to differentiate into stumpy forms, generating high parasitaemia in mice (*Figure 6a*), with no PAD1 positive cells detected 144 hr post infection (*Figure 6b*) and an absence of cell-cycle arrest (*Figure 6c*), contrasting with the development of the parental line. For the other selected targets, we generated conditional knock-down of the corresponding coding genes by RNAi, and for two of them, the hypothetical Tb927.1.4280 (*Figure 6—figure supplement 1*) and the bilobe region protein Tb927.11.15140 (*Figure 6—figure supplement 2*), did not observe a significant effect on the slender to stumpy differentiation in vivo. However, knock-down of hypothetical protein Tb927.4.2750 increased the capacity of the cells to differentiate into stumpy-like cells as judged by their increased sensitivity to the 8-pCPT-cAMP (*Figure 6—figure supplement 3*), whereas NOT5 depletion accelerated the capacity to generate PAD1-positive cells in mice (*Figure 6—figure supplement 4*). These results suggest that these two proteins are 'slender retainers' that need to be inactivated or degraded to allow the cells to efficiently differentiate into stumpy forms. In contrast, TbZC3H20 is a stumpy inducer, whose activity is required for differentiation. In combination, this reveals a role for TbDYRK on both the inhibitory and stimulatory arms of the stumpy formation pathway.

## TbDYRK phosphosite mutation on TbZC3H20 abolishes differentiation competence

To confirm the role of TbDYRK in the regulation of the developmental pathway we used CRISPR mediated allelic mutation to replace the identified TbDYRK phosphosite on TbZC3H20 with a non phosphorylable residue in *T. brucei* EATRO 1125 AnTat1.1 J1339 cells (*Rojas et al., 2019*). A cell line was successfully generated in which one allele of TbZC3H20 was first replaced by the phosphosite mutant (TbZC3H20$^{T283A}$) and the other one then deleted (*Figure 6—figure supplement 5a,b*); this cell line was predicted to be unresponsive to TbDYRK mediated phosphorylation and hence unresponsive to the quorum sensing signal. Growth of the TbZC3H20$^{T283A/-}$ cell line in vivo in comparison with parental phospho-competent TbZC3H20$^{+/+}$ cell lines demonstrated that the mutants were hypervirulent in mice, with the parasites retaining a slender morphology, contrasting with the wild-type parasites or the intermediate cell line containing one mutated allele and one WT allele (TbZC3H20$^{T283A/+}$; *Figure 6—figure supplement 5c*), which became stumpy from day 5 of infection (respectively *Figure 6d* and *Figure 6—figure supplement 5d*). The TbZC3H20$^{T283A/-}$ mutant parasites also did not express PAD1 (*Figure 6e* and additional independent clones- *Figure 6—figure supplement 6*) and did not exhibit cell cycle arrest in G1/G0 (*Figure 6f* and *Figure 6—figure supplement 6c*), confirming their developmental incompetence. This was not related to overall mRNA levels; the TbZC3H20 mRNA levels in the TbZC3H20$^{T283A/-}$ line were approximately equivalent to a distinct single allele replacement line that is differentiation competent (*Figure 6—figure supplement 5b*). Reintroduction of a wild-type TbZC3H20 allele partially restored the WT phenotype in the phosphosite mutant, with cells presenting cell cycle arrest at peak parasitaemia (*Figure 6—figure supplement 7*), although PAD1 expression was not restored. This was potentially related to the multiple rounds of transfection and selection required to generate these lines (*Figure 6—figure supplement 8*) or the contribution of 5'UTR control signals to production of appropriate levels of TbZC3H20 protein.

## Discussion

The quorum-sensing signalling pathway of trypanosomes in their mammalian host provides a framework for evolutionary analysis of signalling in a group separated from the major eukaryotic models (*Keeling and Burki, 2019*). Here, we have analysed through sequence-guided functional analysis and phospho-substrate identification an atypical DYRK component representative of a widely conserved family of proteins centrally implicated in developmental control. This has revealed striking divergence from precedent for this kinase group, particularly an atypical activation mechanism involving unconventional DFS and HxY motifs. It has also defined the first regulatory cascade

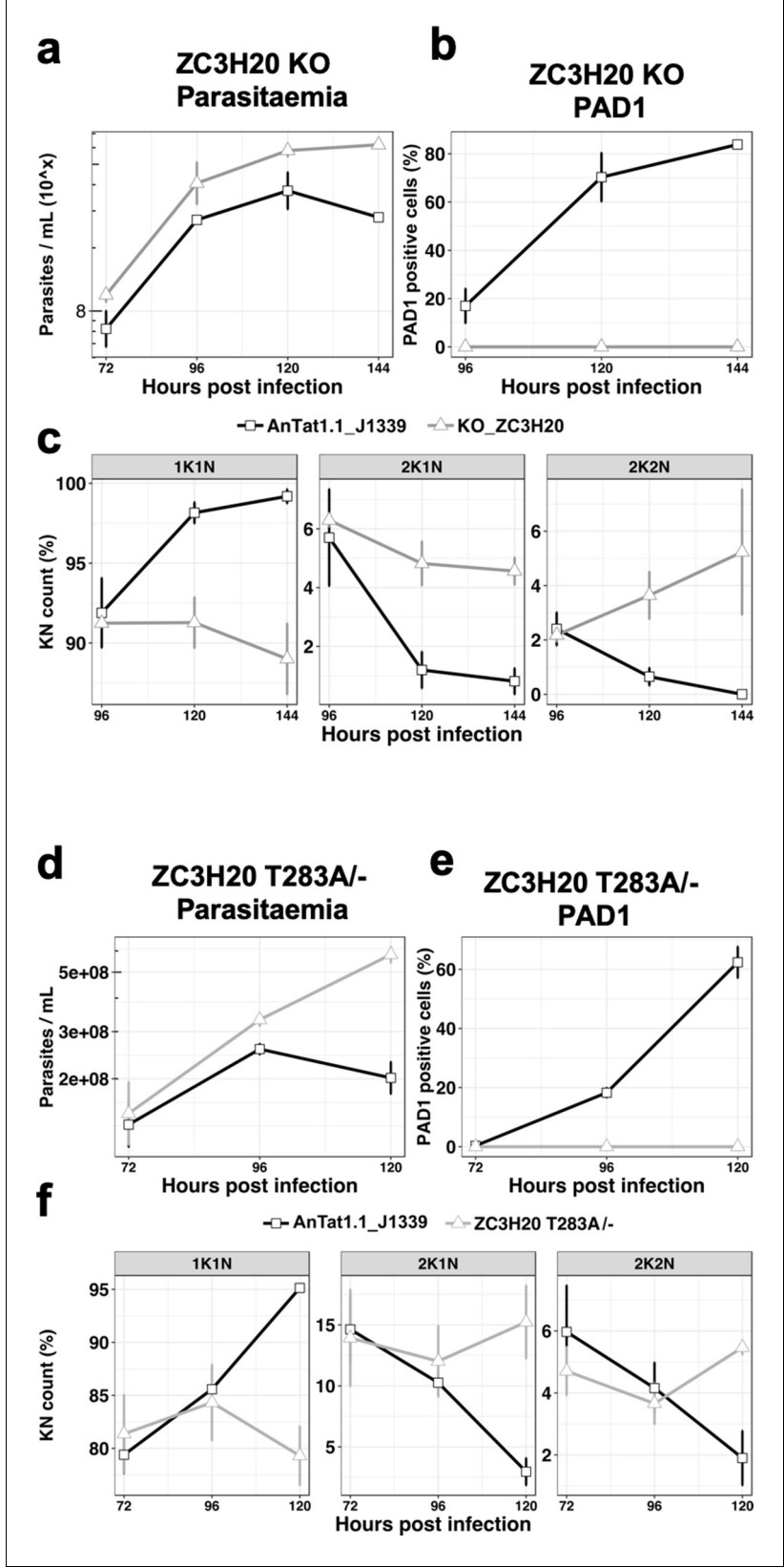

**Figure 6.** The TbDYRK substrate TbZC3H20 is implicated in the slender to stumpy differentiation. (**a**) Parasitaemia of the TbZC3H20 gene KO line (KO_ZC3H20) compared to the parental cell line (AnTat1.1_J1339), n = 3, error bars = SEM. (**b**) Percentage of PAD1 positives cells of the TbZC3H20 gene KO line (KO_ ZC3H20) compared to the

*Figure 6 continued on next page*

*Figure 6 continued*

parental cell line (AnTat1.1_ J1339) on different days of in vivo infection, n = 3, error bars = SEM. (c) Percentage of 1K1N, 2K1N and 2K2N cells on different days of in vivo infection. n = 3, error bars = SEM. (d) Parasitaemia of the TbZC3H20 T283A /- line compared to the parental cell line (AnTat1.1_ J1339), n = 3, error bars = SEM. (e) Percentage of PAD1 positive cells of the TbZC3H20 T283A /- line compared to the parental cell line (AnTat1.1_ J1339), n = 3, error bars = SEM. (f) Percentage of 1K1N, 2K1N and 2K2N cells of the TbZC3H20 T283A /- line compared to the parental cell line (AnTat1.1_ J1339) on different days of in vivo infection, n = 3, error bars = SEM. The online version of this article includes the following figure supplement(s) for figure 6:

**Figure supplement 1.** In vivo phenotypic analysis of the RNAi knocked-down cell line for Tb927.1.4280, uninduced (-dox, black line) or induced with doxycycline at 72 hr post infection (+dox, grey line).
**Figure supplement 2.** In vivo phenotypic analysis of the knocked-down cell line for Tb927.11.15140, uninduced (-dox, black line) or induced with doxycycline at 72 hr post infection (+dox, grey line).
**Figure supplement 3.** In vitro phenotypic analysis of the knocked-down cell line for gene Tb927.4.2750, uninduced (-dox) or induced with doxycycline (+dox), and treated or not with the cell permeable 8'-pCPT-cAMP.
**Figure supplement 4.** In vivo phenotypic analysis of the knocked-down cell line for gene Tb927.3.1920, uninduced (-dox, black line) or induced with doxycycline at 72 hr post-infection (+dox, grey line).
**Figure supplement 5.** Endogenous mutation TbZC3H20 T283A.
**Figure supplement 6.** In vivo phenotypic analysis of three additional clones of the cell line TbZC3H20$^{T283A/-}$ confirms the differentiation resistant phenotype as presented in *Figure 6*.
**Figure supplement 7.** Add-back of a WT copy of TbZC3H20 in the TbZC3H20$^{T283A/-}$ strain in the endogenous context under control of the 3'UTR partially rescues the WT phenotype.
**Figure supplement 8.** Schematic representation of the genetic background of the different cell lines generated and their corresponding number of transfections.
**Figure supplement 9.** Add-back inducible ectopic expression of a WT copy of TbZC3H20 in the TbZC3H20$^{T283A/-}$ strain is leaky and does not rescue the WT phenotype.

directly linking environmental sensing, signal transduction and posttranscriptional regulation in a kinetoplastid parasite.

In other organisms, from yeast to mammals, DYRK kinases have roles, among other cellular functions, in cell differentiation (*Aranda et al., 2011*), stress responses, neural development, myoblast development and embryogenesis, with perturbed DYRK family activity implicated in downs syndrome and some cancers. The involvement of TbDYRK in the control of the trypanosome development, therefore, highlights the evolutionarily conserved involvement of this kinase family in differentiation processes across the diversity of eukaryotic life. Despite this, TbDYRK possesses unique characteristics that suggest kinetoplastid specific features of this kinase distinct from other eukaryotes. Particularly, our analysis of TbDYRK has generated the following model for the activation/function of this molecule (*Figure 7a*). In this model, the kinase would be expressed in slender forms and auto-phosphorylated, possibly in trans, as previously suggested through the action of the divergent NAPA-1 domain (*Han et al., 2012*), on the Y868 of the activation loop during translation, as is typical for this kinase family (*Kinstrie et al., 2010*; *Lochhead et al., 2005*). This auto-phosphorylation at the activation loop, the lack of a second phosphorylable residue on the activation loop (with H in place of tyrosine) and the essential role of the serine the DFS motif (replacing the conventional DFG motif) which is predicted to increase rigidity, suggest this kinase is preserved in a pre-activation state. Indeed, the mutation of this serine to alanine (DFS >DFA) maintained ~70% of kinase activity in vitro, unlike mutation to a glycine which significantly reduced kinase activity. In addition, the modelling of the kinase core shows that the DFS motif is in position 'DFS-in', supporting the probable pre-activation state of the kinase, that would require other mechanisms for activation. For example, the kinase could be activated by phosphorylation on at least three sites in the N-terminal domain (T76, S77, S472) in response to stumpy induction factor in vivo or in response to cell permeable 8-pCPT-cAMP in vitro. The functional analysis also revealed the presence of three inserts in the kinase core of the protein that are essential for kinase function. These may assist binding to partner proteins (*Kelly and Rahmani, 2005*; *Sitz et al., 2004*), similar to human DYRK2, that also acts as an adaptor for the formation of the E3 ubiquitin ligase EDPV (*Maddika and Chen, 2009*). Indeed, the deletion of Insert II, which is cytocidal when cells are exposed to 8-pCPT-cAMP treatment might be caused by effects on the capacity for binding partner interaction.

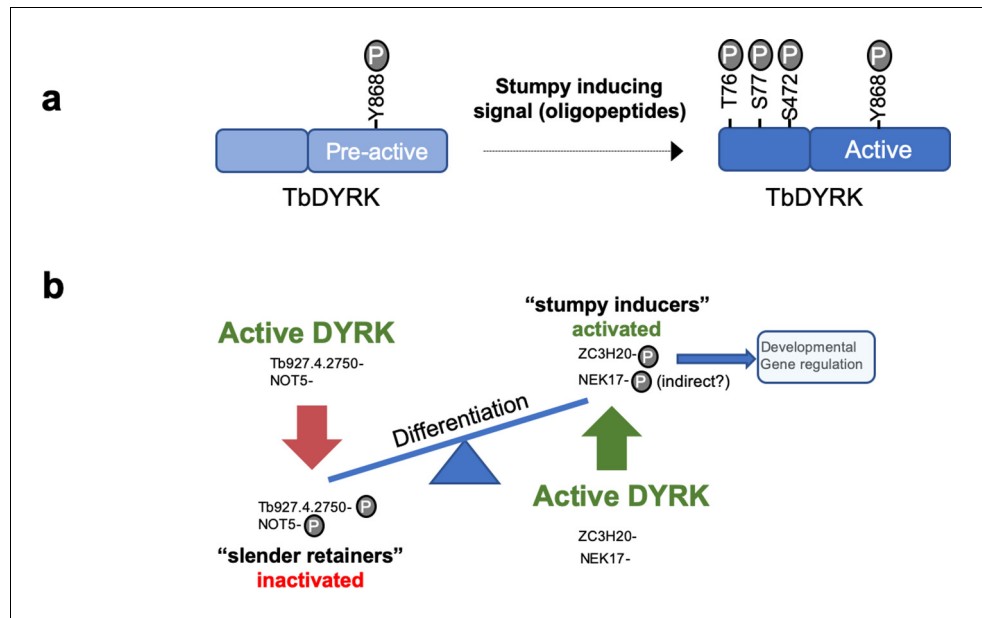

**Figure 7.** Model of the activation mechanisms and function of the TbDYRK. (**a**) Phosphorylation of the pre-active kinase in response to the stumpy inducing signal results in activation of the kinase. (**b**) Consequences of the activation of TbDYRK for the regulation of differentiation, through the inactivation of slender retainer molecules and the activation of stumpy inducers.

We also observed that the ectopic expression of TbDYRK influenced the total level of TbDYRK mRNA derived from the ectopic and the endogenous derived mRNAs, with this control being relaxed when cells exhibited reduced differentiation. This feedback regulation could be mediated via the 3'UTR of the TbDYRK transcript since the ectopic gene, provided with an aldolase 3'UTR, was not subject to the same control. The regulatory effect was not fully correlated with the kinase activity of the expressed TbDYRK mutants but did correspond to the differentiation phenotype generated by the expression of the mutants and so may relate to the activity of other components in the pathway. The combination of kinase activity and/or protein-protein interactions of TbDYRK may control the activity of posttranscriptional regulators, which would then promote differentiation through regulated mRNA stability or translation of downstream targets, but also regulate the level of TbDYRK mRNA itself. Stringent regulation of TbDYRK levels would prevent the parasite irreversibly differentiating to stumpy forms prematurely and is reflected in the very low levels of TbDYRK mRNA and protein detectable in bloodstream forms.

Two complementary quantitative phosphoproteomic approaches to identify substrates of the kinase were performed. First, we compared the phosphoproteome of WT and KO cells and identified around 400 peptides significantly differentially regulated between the cell lines. Interestingly, we observed an enrichment of proteins implicated in chromatin reorganisation and transcriptional regulation, mRNA stability and translation. For example, the histone deacetylase HDAC3 (Tb927.2.2190) presented reduced phosphorylation in the KO cell line, and the mammal DYRK1B has been shown to phosphorylate HDAC5 and 9 to promote myocyte differentiation (**Deng et al., 2005**). A previous study, on the hierarchical organisation of components implicated in the signal transduction of the stumpy differentiation pathway, has identified potential substrates of the MEKK1 kinase, another protein implicated in this process (**McDonald et al., 2018**). Comparing the studies, 12 proteins were shared between the null mutants for each kinase that were less phosphorylated, including the RNA-binding protein, RBP31 (Tb927.4.4230), which has been implicated in decreased translation or mRNA stability (**Lueong et al., 2016**; **De Gaudenzi et al., 2005**), kinetoplastid kinetochore KKT4 protein (Tb927.8.3680) that is essential for chromosomal segregation (**Akiyoshi and Gull, 2014**) and NEK17 (Tb927.10.5950), another component of the QS signalling pathway (**Mony et al., 2014**). The latter protein exhibits reduced phosphorylation on threonine 195 in both

analyses, suggesting that NEK17 phosphorylation requires both TbDYRK and MEKK1 either directly or through the action of another kinase.

Phosphorylation of parasite lysate substrates with the purified active kinase identified 29 proteins that were significantly phosphorylated by the active kinase compared to the inactive mutant. Observed differences between the phosphoproteomic analysis of the null mutant and the lysate analysis are likely due to the technical and biological reasons. Firstly, the use of isobaric tags for analysis of the null and wild type parasites is more sensitive. Secondly, direct and indirect phosphorylation events could contribute to the differences observed. DYRKs are known to act as priming kinases, such that their phosphorylation of substrates is required for further phosphorylation by other kinases, such as GSK3 or PLK (*Gwack et al., 2006*; *Nishi and Lin, 2005*; *Nishi et al., 2008*). This could contribute to the reduction of phosphorylated proteins detected in the null mutant line that are not directly phosphorylated by TbDYRK. Finally, the direct substrate analysis was carried using procyclic form lysates in order to provide sufficient material for analysis, and so stage-specific differences in substrate proteins could contribute. Nonetheless, the comparison of both analyses revealed the presence of 45 common proteins exhibiting a change in phosphorylation regardless of the direction of change or the statistical value, with two proteins that had a statistical change in phosphorylation on the same peptides. The identified molecules showed an enrichment of RNA binding proteins, which is the predicted regulatory level of control for trypanosome development, given the emphasis in kinetoplastid parasites of posttranscriptional regulation. Interestingly, conventional DYRK3 members in other eukaryotes have been identified as regulators of stress granule integrity, with the activity of the kinase influencing the dissolution of these membraneless organelles in order to control the response of cells to stress through regulation of the action of mTOR (*Wippich et al., 2013*; *Rai et al., 2018*). Trypanosome differentiation similarly requires the regulation of TOR activity and the control of development through the dynamic association of predicted RNA binding proteins with stress granules, and TbDYRK - with features of DYRK2 and DYRK3 families - may functionally connect these regulatory components of trypanosome cell cycle exit and stumpy formation.

Genetic validation of some of the identified proteins in the stumpy differentiation pathway, by knock-out or conditional knock-down, revealed that TbDYRK acts on both the inhibitory and stimulatory arms of the differentiation control pathway (*Figure 7b*). Indeed, knock-down of the hypothetical protein Tb927.4.2750 rendered the cells more sensitive to 8-pCPT-cAMP and knock-down of NOT5 led to an increase in cell differentiation at a lower parasitaemia. These observations suggest that the phosphorylation of these proteins by TbDYRK would inactivate their function, allowing the cells to differentiate. This identifies these proteins as so called 'slender retainers' that are required for the cells to remain as proliferative bloodstream forms. Conversely, the knock-out of the zinc finger protein TbZC3H20 rendered cells unable to differentiate from slender to stumpy forms, indicating that its phosphorylation by TbDYRK would activate the protein to promote differentiation, a so-called 'stumpy inducer'. Consistent with this, the deletion of TbZC3H20 rendered the cells unable to arrest in G1/G0 and this phenotype was reproduced when only the phosphosite differentially detected in TbDYRK null mutants was mutated. Although this supports a direct relationship between TbZC3H20 phosphorylation capacity and developmental competence, we found that re-expression of a WT copy of TbZC3H20, either by inducible ectopic expression (*Figure 6—figure supplement 9*) or in its endogenous context (*Figure 6—figure supplement 7*), only partially restored the WT phenotype. For inducible expression, we observed TbZC3H20 ectopic expression in both induced and uninduced samples; this has the potential to select for the loss of stumpy formation capacity because differentiation regulators can generate irreversible cell cycle arrest if inappropriately expressed in slender forms (*McDonald et al., 2018*). Similarly, the multiple rounds of transfection and selection associated with reintroduction of the wild type allele into cell lines where the endogenous gene copies are replaced with the phosphosite mutant can select for reduced pleomorphism. In consequence, our biochemical and gene replacement data support a direct association of TbZC3H20 T283 phosphorylation with TbDYRK activity, but experimental challenges prevent a definitive link being fully established.

In combination this study positions TbDYRK as a pivotal regulator of the trypanosome QS sensing pathway, acting on both differentiation activators and inhibitors through phosphorylation mediated control. Moreover, its effects on TbZC3H20 link the signal transduction cascade to post transcriptional regulation in the parasite, mRNAs regulated by this molecule being important in the parasite's developmental events (*Ling et al., 2011*; *Liu et al., 2020*).

# Materials and methods

**Key resources table**

| Reagent type (species) or resource | Designation | Source or reference | Identifiers | Additional information |
|---|---|---|---|---|
| Antibody | Anti-digoxinegin-AP Fab fragments | Roche | Cat# 11093274910 | Sheep polyclonal, dilution according to manufacturer instruction |
| Antibody | Anti-Ty1 epitope tag specific BB2 antibody | *Bastin et al., 1996*; hybridoma cell line a gift of Keith Gull, Oxford University/ available through Thermofisher | Cat#MA5-23513; RRID:AB_2610644 | Hybrydome mouse monoclonal, clone BB2, WB (1:20), IF (1:5) |
| Antibody | Anti-Mouse AlexaFluor 488 | molecular probes | Cat# A1101 | Goat polyclonal, IF (1:500) |
| Antibody | Anti-Mouse AlexaFluor 568 | abcam | Cat# ab175701 | Goat polyclonal, IF (1:500) |
| Antibody | Anti-Rabbit AlexaFluor 488 | invitrogen | Cat# A1108 | Goat polyclonal, IF (1:500) |
| Antibody | Anti-Rabbit AlexaFluor 568 | invitrogen | Cat# A11036 | Goat polyclonal, IF (1:500) |
| Antibody | IRDye 680 anti-mouse secondary antibody | Li-Cor | Cat#P/N 925-68070; RRID:AB_2651128 | Goat polyclonal, IF (1:5000) |
| Antibody | IRDye 800CW anti-Mouse IgG (H + L) secondary antibody | Li-Cor | Cat#P/N 925-32210; RRID:AB_2687825 | Goat polyclonal, IF (1:5000) |
| Antibody | Anti-PAD1 | *Dean et al., 2009* | N/A | Rabbit polyclonal, WB (1:1000), IF (1:1000) |
| Antibody | Anti-Thio-phosphate ester [51-8] | abcam | Cat# ab92570 | Rabbit monoclonal, clone 51-8, WB (1:1000) |
| Cell line (*Spodoptera frugiperda*) | SF9 insect cells | gibco | Cat# 12659017 | |
| Cell line (*Trypanosoma brucei*) | *Trypanosoma brucei EATRO 1125 AnTat1.1 90:13* | *Engstler and Boshart, 2004* | N/A | |
| Cell line (*Trypanosoma brucei*) | *Trypanosoma brucei EATRO 1125 AnTat1.1 J1339* | *Rojas et al., 2019* | N/A | |
| Commercial assay, kit | DIG RNA labelling kit (SP6/T7) | Roche | Cat# 11175025910 | |
| Commercial assay, kit | Dneasy Blood & Tissue kit | Qiagen | Cat# 69506 | |
| Commercial assay, kit | Luna Universal qPCR master mix | NEB | Cat# M3003L | |
| Commercial assay, kit | Monarch DNA Gel extraction kit | NEB | Cat# T1020L | |
| Commercial assay, kit | Monarch PCR DNA Cleanup kit | NEB | Cat# T1030S | |
| Commercial assay, kit | Quickchange II site directed mutagenesis kit | Agilent | Cat#200523 | |
| Commercial assay, kit | Rapid DNA ligation kit | invitrogen | Cat# K1423 | |
| Commercial assay, kit | RNeasy mini kit | Qiagen | Cat# 74106 | |

*Continued on next page*

*Continued*

| Reagent type (species) or resource | Designation | Source or reference | Identifiers | Additional information |
|---|---|---|---|---|
| Commercial assay, kit | Zero blunt TOPO PCR cloning kit | invitrogen | Cat# 45-0245 | |
| Gene (*Trypanosoma brucei*) | bilobe region protein, putative | *Hu et al., 2015* | Tb927.11.15140 | |
| Gene (*Trypanosoma brucei*) | hypothetical protein | N/A | Tb927.1.4280 | |
| Gene (*Trypanosoma brucei*) | hypothetical protein | N/A | Tb927.4.2750 | |
| Gene (*Trypanosoma brucei*) | Not5 | *Schwede et al., 2008* | Tb927.3.1920 | |
| Gene (*Trypanosoma brucei*) | TbDYRK (formely YAK) | *Mony et al., 2014* | Tb927.10.15020 | |
| Gene (*Trypanosoma brucei*) | ZC3H20 | *Liu et al., 2020* | Tb927.7.2660 | |
| Other | [g-32P]-ATP (3000 Ci/mmol) | Perkin Elmer | Cat# NEG502A250UC | |
| Other | 8-(4-Chlorophenylthio) adenosine 3',5'-cyclic monophosphate sodium salt | Sigma-aldrich | Cat# C3912 | |
| Other | Amaxa basic parasite nucleofector kit 2 solution | Lonza | Cat#VMI-1021 | |
| Other | ATP-gamma-S | Abcam | Cat# ab18911 | |
| Other | CDP star | Roche | Cat# 11685627001 | |
| Other | Cellfectin II Reagent | gibco | Cat# 10362100 | |
| Other | DE52 | Whatman | Cat# 4057200 | |
| Other | Dyneabeads protein G | Invitrogen | Cat# 10003D | |
| Other | Grace's Medium | gibco | Cat# 11595-030 | |
| Other | HMI-9 Medium | Life Technologies | Cat#074-90915 | |
| Other | MOPS | sigma aldrich | Cat# M3183 | |
| Other | NuPAGE 4-12% Bis-Tris Gel | invitrogen | Cat# NP0322BOX | |
| Other | NuPAGE MES SDS running buffer (20x) | invitrogen | Cat# NP0002 | |
| Other | Oligo dT(20) Primer | Invitrogen | Cat# 18418020 | |
| Other | Paraformaldehyde | sigma aldrich | Cat# P6148 | |
| Other | Phire green hot start II DNA polymerase | thermo scientific | Cat# F-124S | |
| Other | PNBM (p-Nitrobenzyl mesylate) in DMSO | abcam | Cat# ab138910 | |
| Other | ProLong Diamond antifade mount | invitrogen | Cat# P36965 | |
| Other | Protease inhibitor cocktail completme EDTA-free | Roche | Cat# 11873580001 | |
| Other | RNAse H | NEB | Cat# M0297S | |
| Other | SDM-79 medium | Life Technologies | Cat#074-90916 | |
| Other | Sf-900 1.3X | gibco | Cat# 10967-032 | |
| Other | Sf-900 II SFM medium | gibco | Cat# 10902-096 | |

*Continued*

| Reagent type (species) or resource | Designation | Source or reference | Identifiers | Additional information |
|---|---|---|---|---|
| Other | Superscript III reverse transcriptase | Invitrogen | Cat# 18080-044 | |
| Peptide, recombinant protein | Beta-Casein | Sigma-aldrich | Cat# C6905-250MG | |
| Peptide, recombinant protein | Casein-dephosphorylated | Sigma-aldrich | Cat# C4032-100MG | |
| Peptide, recombinant protein | Caspase 9, 200 first amino acids | Cloud-clone corp. | Cat# RPA627Mu01 | |
| Peptide, recombinant protein | Histone H1 | Milipore | Cat# 14-155 | |
| Peptide, recombinant protein | Histones cores H2A, 2B, 3, 4 | BioVision | Cat# 7677-50 | |
| Peptide, recombinant protein | MBP-depho sphorylates | Millipore | Cat# 13-110 | |
| Recombinant DNA reagent | pALC14 | *Pusnik et al., 2007* | N/A | |
| Recombinant DNA reagent | pALC14_1.4280 | This study | N/A | Derived from pALC14 from *Pusnik et al. (2007)*, generated in Keith Matthews's Lab |
| Recombinant DNA reagent | pALC14_NOT5 | This study | N/A | Derived from pALC14 from *Pusnik et al. (2007)*, generated in Keith Matthews's Lab |
| Recombinant DNA reagent | pALC14_Tb927.11.15140 | This study | N/A | Derived from pALC14 from *Pusnik et al. (2007)*, generated in Keith Matthews's Lab |
| RecombinantDNA reagent | pALC14_Tb927.4.2750 | This study | N/A | Derived from pALC14 from *Pusnik et al. (2007)*, generated in Keith Matthews's Lab |
| Recombinant DNA reagent | pDEX577-Y | *Kelly et al., 2007* | N/A | |
| Recombinant DNA reagent | pDEX577-Y_TbDYRK_ΔI::YFP-TY | This study | N/A | Derived from pDEX577-Y from *Kelly et al. (2007)*, generated in Keith Matthews's Lab |
| Recombinant DNA reagent | pDEX577-Y_TbDYRK_ΔII::YFP-TY | This study | N/A | Derived from pDEX577-Y from *Kelly et al. (2007)*, generated in Keith Matthews's Lab |
| Recombinant DNA reagent | pDEX577-Y_TbDYRK_ΔIII::YFP-TY | This study | N/A | Derived from pDEX577-Y from *Kelly et al. (2007)*, generated in Keith Matthews's Lab |
| Recombinant DNA reagent | pDEX577-Y_TbDYRK_H866A_Y868F::YFP-TY | This study | N/A | Derived from pDEX577-Y from *Kelly et al. (2007)*, generated in Keith Matthews's Lab |

*Continued*

| Reagent type (species) or resource | Designation | Source or reference | Identifiers | Additional information |
|---|---|---|---|---|
| Recombinant DNA reagent | pDEX577-Y_TbDYRK_H866A::YFP-TY | This study | N/A | Derived from pDEX577-Y from *Kelly et al. (2007)*, generated in Keith Matthews's Lab |
| Recombinant DNA reagent | pDEX577-Y_TbDYRK_K592A::YFP-TY | This study | N/A | Derived from pDEX577-Y from *Kelly et al. (2007)*, generated in Keith Matthews's Lab |
| Recombinant DNA reagent | pDEX577-Y_TbDYRK_NM::YFP-TY | This study | N/A | Derived from pDEX577-Y from *Kelly et al. (2007)*, generated in Keith Matthews's Lab |
| Recombinant DNA reagent | pDEX577-Y_TbDYRK_S472A::YFP-TY | This study | N/A | Derived from pDEX577-Y from *Kelly et al. (2007)*, generated in Keith Matthews's Lab |
| Recombinant DNA reagent | pDEX577-Y_TbDYRK_S77A::YFP-TY | This study | N/A | Derived from pDEX577-Y from *Kelly et al. (2007)*, generated in Keith Matthews's Lab |
| Recombinant DNA reagent | pDEX577-Y_TbDYRK_S856A::YFP-TY | This study | N/A | Derived from pDEX577-Y from *Kelly et al. (2007)*, generated in Keith Matthews's Lab |
| Recombinant DNA reagent | pDEX577-Y_TbDYRK_S856G::YFP-TY | This study | N/A | Derived from pDEX577-Y from *Kelly et al. (2007)*, generated in Keith Matthews's Lab |
| Recombinant DNA reagent | pDEX577-Y_TbDYRK_T76A::YFP-TY | This study | N/A | Derived from pDEX577-Y from *Kelly et al. (2007)*, generated in Keith Matthews's Lab |
| Recombinant DNA reagent | pDEX577-Y_TbDYRK_Y868F::YFP-TY | This study | N/A | Derived from pDEX577-Y from *Kelly et al. (2007)*, generated in Keith Matthews's Lab |
| Recombinant DNA reagent | pDEX577-Y_ZC3H20::YFP-TY | This study | N/A | Derived from pDEX577-Y from *Kelly et al. (2007)*, generated in Keith Matthews's Lab |
| Recombinant DNA reagent | pEnT6B-Y | *Kelly et al., 2007* | N/A | |
| Recombinant DNA reagent | pEnT6B-Y_UTRs_TbDYRK | This study | N/A | Derived from pEnT6B-Y from *Kelly et al. (2007)*, generated in Keith Matthews's Lab |
| Recombinant DNA reagent | pEnT6P-Y | *Kelly et al., 2007* | N/A | Derived from pEnT6B-Y from *Kelly et al. (2007)*, generated in Keith Matthews's Lab |
| Recombinant DNA reagent | pEnT6P-Y_UTRs_TbDYRK | This study | N/A | Derived from pEnT6B-Y from *Kelly et al. (2007)*, generated in Keith Matthews's Lab |
| Recombinant DNA reagent | pET28a | Novagen | Cat#69864 | |
| Recombinant DNA reagent | pFASTBac1 expression vector | Invitrogen | Cat# 10360-010 | |

*Continued on next page*

*Continued*

| Reagent type (species) or resource | Designation | Source or reference | Identifiers | Additional information |
|---|---|---|---|---|
| Recombinant DNA reagent | pFASTBac1_TY-YFP-TY | This study | N/A | Derived from pFASTBac1 from Novagen, generated in Keith Matthews's Lab |
| Recombinant DNA reagent | pFASTBac1_TY-YFP-TY::TbDYRK_ΔI | This study | N/A | Derived from pFASTBac1 from Novagen, generated in Keith Matthews's Lab |
| Recombinant DNA reagent | pFASTBac1_TY-YFP-TY::TbDYRK_ΔII | This study | N/A | Derived from pFASTBac1 from Novagen, generated in Keith Matthews's Lab |
| Recombinant DNA reagent | pFASTBac1_TY-YFP-TY::TbDYRK_ΔIII | This study | N/A | Derived from pFASTBac1 from Novagen, generated in Keith Matthews's Lab |
| Recombinant DNA reagent | pFASTBac1_TY-YFP-TY::TbDYRK_H866A | This study | N/A | Derived from pFASTBac1 from Novagen |
| Recombinant DNA reagent | pFASTBac1_TY-YFP-TY::TbDYRK_H866A_Y868F | This study | N/A | Derived from pFASTBac1 from Novagen, generated in Keith Matthews's Lab |
| Recombinant DNA reagent | pFASTBac1_TY-YFP-TY::TbDYRK_K592A | This study | N/A | Derived from pFASTBac1 from Novagen, generated in Keith Matthews's Lab |
| Recombinant DNA reagent | pFASTBac1_TY-YFP-TY::TbDYRK_NM | This study | N/A | Derived from pFASTBac1 from Novagen, generated in Keith Matthews's Lab |
| Recombinant DNA reagent | pFASTBac1_TY-YFP-TY::TbDYRK_S472A | This study | N/A | Derived from pFASTBac1 from Novagen, generated in Keith Matthews's Lab |
| Recombinant DNA reagent | pFASTBac1_TY-YFP-TY::TbDYRK_S77A | This study | N/A | Derived from pFASTBac1 from Novagen, generated in Keith Matthews's Lab |
| Recombinant DNA reagent | pFASTBac1_TY-YFP-TY::TbDYRK_S856A | This study | N/A | Derived from pFASTBac1 from Novagen, generated in Keith Matthews's Lab |
| Recombinant DNA reagent | pFASTBac1_TY-YFP-TY::TbDYRK_S856G | This study | N/A | Derived from pFASTBac1 from Novagen, generated in Keith Matthews's Lab |
| Recombinant DNA reagent | pFASTBac1_TY-YFP-TY::TbDYRK_T76A | This study | N/A | Derived from pFASTBac1 from Novagen, generated in Keith Matthews's Lab |
| Recombinant DNA reagent | pFASTBac1_TY-YFP-TY::TbDYRK_Y868F | This study | N/A | Derived from pFASTBac1 from Novagen, generated in Keith Matthews's Lab |
| Recombinant DNA reagent | pGEMTeasy | Promega | Cat# A1360 | |
| Recombinant DNA reagent | pJ1339 | Kindly provided by Dr Jack Sunter, Oxford Brookes University | N/A | |
| Recombinant DNA reagent | pPOTv6 | Kindly provided by Dr Sam Dean, Oxford University | N/A | |
| Recombinant DNA reagent | pPOTv6_ZC3 H20_T283A | This study | N/A | Derived from pPOTv6, generated in Keith Matthews's Lab |

*Continued on next page*

*Continued*

| Reagent type (species) or resource | Designation | Source or reference | Identifiers | Additional information |
|---|---|---|---|---|
| Recombinant DNA reagent | pPOTv7 | Kindly provided by Dr Sam Dean, Oxford University | N/A | |
| Recombinant DNA reagent | pTOPO PCR blunt II | invitrogen | Cat# 45-0245 | |
| Recombinant DNA reagent | pPOTv6_BLEO_BSD | This study | N/A | Derived from pPOTv6, generated in Keith Matthews's Lab |
| Recombinant DNA reagent | pPOTv6_BLEO_ TY::ZC3H20_BSD | This study | N/A | Derived from pPOTv6, generated in Keith Matthews's Lab |
| Software, algorithm | Bioconductor | Bioconductor | http://www.bioconductor.org/ | |
| Software, algorithm | Blast | NCBI | https://blast.ncbi.nlm.nih.gov/Blast.cgi | |
| Software, algorithm | ClustalXv2 | *Larkin et al., 2007* | http://www.clustal.org/clustal2/ | |
| Software, algorithm | Eukaryotic Linear Motif resource | *Gouw et al., 2018* | http://elm.eu.org/infos/about.html | |
| Software, algorithm | GeneDB | *Hertz-Fowler, 2004* | http://www.genedb.org/Homepage | |
| Software, algorithm | iTASSER | *Yang and Zhang, 2015* | https://zhanglab.ccmb.med.umich.edu/I-TASSER/ | |
| Software, algorithm | Jalview | *Waterhouse et al., 2009* | N/A | |
| Software, algorithm | LeishGEdit | *Beneke et al., 2017* | http://www.leishgedit.net/Home.html | |
| Software, algorithm | MEGA7 | *Kumar et al., 2016* | https://www.megasoftware.net | |
| Software, algorithm | MUSCLE | *Madeira et al., 2019* | https://www.ebi.ac.uk/Tools/msa/muscle/ | |
| Software, algorithm | OrthoMCL | *Chen et al., 2006* | http://orthomcl.org/orthomcl/ | |
| Software, algorithm | Pfam | *El-Gebali et al., 2019* | http://pfam.xfam.org | |
| Software, algorithm | PRATT v2.1 | *Jonassen et al., 1995* | N/A | |
| Software, algorithm | PyMol1.8 | Schrödinger | https://pymol.org/2/ | |
| Software, algorithm | R | R | https://www.r-project.org/ | |
| Software, algorithm | RNAit | *Redmond et al., 2003* | https://dag.compbio.dundee.ac.uk/RNAit/ | |
| Software, algorithm | Rstudio | Rstudio | https://rstudio.com/ | |
| Software, algorithm | TriTrypDB | *Aslett et al., 2010* | http://tritrypdb.org/tritrypdb/ | |
| Strain, strain background (*Escherichia coli*) | MAX Efficiency DH10Bac chemically competent cells | gibco | Cat# 10361012 | |

*Continued on next page*

*Continued*

| Reagent type (species) or resource | Designation | Source or reference | Identifiers | Additional information |
|---|---|---|---|---|
| Strain, strain background (*Mus musculus*) | Mouse MF1, female | Charles River | N/A | |
| Transfected construct | *Trypanosoma brucei EATRO 1125 AnTat1.1 90:13_RNAi_Not5* | This study | N/A | Cell lines generated in Keith Matthews's Lab |
| Transfected construct | *Trypanosoma brucei EATRO 1125 AnTat1.1 90: 13_RNAi_Tb927.1.4280* | This study | N/A | Cell lines generated in Keith Matthews's Lab |
| Transfected construct | *Trypanosoma brucei EATRO 1125 AnTat1.1 90:13_RNAi_ Tb927.11.15140* | This study | N/A | Cell lines generated in Keith Matthews's Lab |
| Transfected construct | *Trypanosoma brucei EATRO 1125 AnTat1.1 90: 13_RNAi_Tb927.4.2750* | This study | N/A | Cell lines generated in Keith Matthews's Lab |
| Transfected construct | *Trypanosoma brucei EATRO 1125 AnTat1.1 90 :13_TbDYRK_ΔI* | This study | N/A | Cell lines generated in Keith Matthews's Lab |
| Transfected construct | *Trypanosoma brucei EATRO 1125 AnTat1.1 90:13_ TbDYRK_ΔII* | This study | N/A | Cell lines generated in Keith Matthews's Lab |
| Transfected construct | *Trypanosoma brucei EATRO 1125 AnTat1.1 90:13_ TbDYRK_ΔIII* | This study | N/A | Cell lines generated in Keith Matthews's Lab |
| Transfected construct | *Trypanosoma brucei EATRO 1125 AnTat1.1 90:13_ TbDYRK_H866A* | This study | N/A | Cell lines generated in Keith Matthews's Lab |
| Transfected construct | *Trypanosoma brucei EATRO 1125 AnTat1.1 90:13_Tb DYRK_H866A_Y868F* | This study | N/A | Cell lines generated in Keith Matthews's Lab |
| Transfected construct | *Trypanosoma brucei EATRO 1125 AnTat1.1 90:13_ TbDYRK_K592A* | This study | N/A | Cell lines generated in Keith Matthews's Lab |
| Transfected construct | *Trypanosoma brucei EATRO 1125 AnTat1.1 90:13_ TbDYRK_KO* | This study | N/A | Cell lines generated in Keith Matthews's Lab |
| Transfected construct | *Trypanosoma brucei EATRO 1125 AnTat1.1 90:13_ TbDYRK_NM* | This study | N/A | Cell lines generated in Keith Matthews's Lab |
| Transfected construct | *Trypanosoma brucei EATRO 1125 AnTat1.1 90:13_ TbDYRK_S472A* | This study | N/A | Cell lines generated in Keith Matthews's Lab |

*Continued on next page*

*Continued*

| Reagent type (species) or resource | Designation | Source or reference | Identifiers | Additional information |
|---|---|---|---|---|
| Transfected construct | *Trypanosoma brucei* EATRO 1125 AnTat1.1 90:13_ TbDYRK_S77A | This study | N/A | Cell lines generated in Keith Matthews's Lab |
| Transfected construct | *Trypanosoma brucei* EATRO 1125 AnTat1.1 90:13_ TbDYRK_S856A | This study | N/A | Cell lines generated in Keith Matthews's Lab |
| Transfected construct | *Trypanosoma brucei* EATRO 1125 AnTat1.1 90:13_  TbDYRK_S856G | This study | N/A | Cell lines generated in Keith Matthews's Lab |
| Transfected construct | *Trypanosoma brucei* EATRO 1125 AnTat1.1 90:13_ TbDYRK_T76A | This study | N/A | Cell lines generated in Keith Matthews's Lab |
| Transfected construct | *Trypanosoma brucei* EATRO 1125 AnTat1.1 90:13_ TbDYRK_Y868F | This study | N/A | Cell lines generated in Keith Matthews's Lab |
| Transfected construct | *Trypanosoma brucei* EATRO 1125 AnTat1.1 J1339 _ZC3H20_KO | This study | N/A | Cell lines generated in Keith Matthews's Lab |
| Transfected construct | *Trypanosoma brucei* EATRO 1125 AnTat1.1 J1339_ ZC3H20_T283A/- | This study | N/A | Cell lines generated in Keith Matthews's Lab |
| Transfected construct | *Trypanosoma brucei* EATRO 1125 AnTat1.1 J1339_ZC3H20 _T283A/-_ pDEX577-Y-ZC3H20 | This study | N/A | Cell lines generated in Keith Matthews's Lab |

## Ethics statement

Animal experiments in this work were carried out in accordance with the local ethical approval requirements of the University of Edinburgh and the UK Home Office Animal (Scientific Procedures) Act (1986) under licence number 60/4373.

## Trypanosome culture, constructs and transfection

Pleomorphic *T. brucei brucei* EATRO 1125 AnTat1.1 90:13 cells (*Engstler and Boshart, 2004*) were used for the phenotypic analysis of the ectopic expression of mutants of the DYRK kinase and knock-down of the potential substrates in bloodstream forms. The *T. brucei brucei* EATRO 1125 AnTat1.1 90:13 cell line expresses the T7 RNA polymerase and tetracycline repressor protein. Pleomorphic *T. brucei brucei* EATRO 1125 AnTat1.1 cells were transfected by the plasmid J1339 (kindly provided by Dr. Jack Sunter), that contains the T7 RNA polymerase and the CRISPR/Cas9.

Slender forms were either grown in-vitro or harvested from MF1 female mice at 3 days post infection. Stumpy forms were harvested from MF1 female mice between 5- and 7 days post-infection. All bloodstream form cell lines were grown in vitro in HMI-9 at 37°C in 5% $CO_2$. In vitro differentiation into 'stumpy-like' forms was performed for 96 hr using 100 µM of the cell permeable cyclic AMP analogue, 8-pCPT-cAMP, purchased from Sigma-Aldrich (United Kingdom) (*Mony et al., 2014*).

Ectopic expression constructs were generated using the pDex577-Y vector, that integrates in to the 177 bp repeat mini-chromosome region. The different bloodstream RNAi cell lines were

generated using the stem loop vector pALC14 (*Pusnik et al., 2007*). Both ectopic expression and knock down were initiated by the addition of 2 µg/mL of doxycycline in the culture medium.

Null mutant constructs of the gene TbDYRK were created by replacing the YFP and TY tags of the pEnT6B-Y and pEnT6P-Y vectors (*Kelly et al., 2007*) with fragments of the 3' and 5'UTRs of the target gene then integrated in the genome, as describe in *McDonald et al. (2018)*.

CRISPR/Cas9 knock-out construct of Tb927.7.2660 was generated as described in *Beneke et al. (2017)* using the pPOTv7 plasmid and transfected in the *T. brucei* EATRO 1125 AnTat1.1 J1339 strain. Correct integration of the construct and the deletion of the targeted gene were verified with the pairs of primers MC016/018 + MC017/018 and MC016/075, respectively.

Endogenous mutation of the gene coding for the protein TbZC3H20 was generated, in the J1339 strain, by cloning the mutated version (OL068/069) of the gene into the pPOTv6-BSD plasmid using primers OL066/067 and the restriction enzymes HindIII/ScaI, replacing the tagging cassette. Integration was then performed as described in *Beneke et al. (2017)*. The second allele was deleted using the pPOTv7-HYG plasmid as described in *Beneke et al. (2017)* (Primer list in *Supplementary file 1*). The endogenous add-back of the WT version of the gene coding for the protein TbZC3H20 was generated by cloning the WT version of the gene into pPOTv6-BLEO-BSD in place of the mNG tag. Final transfected template fragment was amplified using the primers OL062/082. Correct mutagenesis and deletion were assessed by sequencing (Primer list in *Supplementary file 1*).

Pleomorph transfections were performed as described by *MacGregor et al. (2013)*. Selection was applied by using the appropriate drugs: Geneticin (G418, 90:13 = 2.5 µg/ml), Hygromycin (HYG, All strains = 0.5 µg/ml), Puromycin (PURO, 90:13 = 0.25 µg/ml, J1339 = 0.05 µg/ml), Blasticidin (BSD, 90:13 = 10 µg/ml, J1339 = 2 µg/ml) and Phleomycin (BLE, 90:13 = 2.5 µg/ml).

## pDEX577-Y

pDEX577_Tb927.10.15020::YFP-TY was generated by Gibson cloning as described in *McDonald et al. (2018)*. Site direct mutagenesis (Primer list in *Supplementary file 1*) was then performed to generate the different mutants and their fidelity confirmed by sequencing (Primer list in *Supplementary file 1*). Plasmids were linearized with NotI, prior to transfection into the *T. brucei* EATRO 1125 AnTat1.1 90:13 strain.

## pALC14

RNAi target gene fragments were selected based on the default settings of the RNAit software (*Redmond et al., 2003*). Fragments were amplified using the pairs of primers indicated in the primer list in *Supplementary file 1* and cloned into pCR-Blunt II-TOPO (Invitrogen), prior to sequencing. Resulting constructs were then digested by HindIII/XbaI and the extracted fragments cloned into pALC14 plasmid opened by the same enzymes. The second round of cloning was performed by digestion by BamHI/XhoI, allowing the head-to-head arrangement of the two identical fragments into the pALC14, generating the stem loop (*Pusnik et al., 2007*). Final plasmids were linearized with NotI, prior to transfection into the *T. brucei* AnTat1.1 90:13 strain.

## pFASTBac1_TY-YFP-TY::Tb927.10.15020

Amplification of TY-YFP from pDEX577-Y was performed using the primer pair MC003/004 and cloned into pGEMTeasy (Promega), prior to sequencing (MC007/MC015). Next, the PstI restriction site into the YFP gene was removed by site directed mutagenesis using MC043/044 and the construct was linearised by the enzymes BssHII/PstI to allow the ligation of the hybridised primers MC005/006, for the integration of the second TY tag. The final tag TY-YFP-TY was then extracted from pGEMTeasy by BamHI/PstI and cloned into the pFASTBac1 vector (Gibco) previously digested by the same set of enzymes, to generate the pFASTBac1_TY-YFP-TY plasmid, allowing tagging in N-terminal or C-terminal. Cloning at the N-terminal end was performed by amplification of the gene TbDYRK with the primers MC001/002, subcloned into pGEMTeasy for sequencing (MC007/010/011/012/013/014/015), and digestion by BssHII/PstI to generate the final pFASTBac1_TY-YFP-TY::Tb927.10.15020. This final plasmid was then transformed into Bac10 bacteria (Gibco) to generate the baculovirus, necessary for the infection of insect cells, according to manufacturer instructions.

Site direct mutagenesis were then performed to generate the different mutants and then verified by sequencing.

## Protein expression and purification

Expression and purification of the recombinant tagged TY-YFP-TY:: TbDYRK was performed using the SF9 insect cell (Gibco) in SF900 II serum free medium, according to manufacturer's instructions. Lysis of infected SF9, by the baculovirus allowing expression of the kinase, was performed with RIPA buffer (25 mM Tris pH7-8, 150 mM NaCl, 0.1%SDS, 0.5% Sodium deoxycholate, 1% Triton X-100, Protease inhibitor cocktail Roche mini EDTA-free, Benzonase at 25 U/mL) and incubated 30 min on ice. After sonication, samples were clarified and the supernatant incubated with the αBB2 antibody (*Bastin et al., 1996*). Immunoprecipitation was then performed using the magnetic Dyneabeads protein G (Invitrogen), according to manufacturer's instructions. Elution was carried out with 25 mM Tris pH7.4.

## Immunofluorescence

Cell cycle analysis and PAD1 protein expression analysis were carried out by staining ice-cold methanol fixed cells with 4',6-diamidino-2-phenylindole (DAPI) (100 ng/ ml) and an anti-PAD1 antibody as previously described (*MacGregor and Matthews, 2012*).

## Protein visualisation and western blotting

Protein samples were boiled for 5 min in Laemmli loading buffer (except PAD1 samples, that remained unboiled), separated by SDS–PAGE (NuPAGE gel 4–12% Bis-Tris, Invitrogen) and visualized either by Coomassie staining or SYPRO Ruby Protein Gel Stain (Invitrogen) using a Typhoon 9400 scanner (Amersham Biosciences) with lex = 457 nm and lem = 610 nm. Alternatively, proteins were separated by SDS–PAGE on NuPAGE 4–12% Bis-Tris gels and blotted onto polyvinylidene difluoride (PVDF) membranes (Pierce). After blocking with Odyssey Blocking buffer for at least 30 min at room temperature (RT), membranes were incubated with antibodies 1- to 3 hr at RT or overnight at 4°C under agitation in 2% BSA in TBS-T (0.1% Tween in TBS). The primary antibodies were used at the following dilutions: αPAD1 (1:1000); αBB2 (1:5, [*Bastin et al., 1996*]); αThioP (1:1000, Thio-phosphate ester [51-8], Abcam). After three washes in TBS-T, proteins were visualised by incubating the membrane for 1 hr at RT with a secondary antibody conjugated to a fluorescent dye diluted 1:5000 in 50%Odyssey Blocking buffer/50% TBS-T. Finally, membranes were scanned using a LI-COR Odyssey imager system.

## Northern blotting and transcriptome analysis

For northern blotting, RNA preparation and analysis were carried out as described by *McDonald et al. (2018)*. Briefly, the targeting sequence was chosen using the RNAit website (*Redmond et al., 2003*), cloned into pGEMTeasy vector and then subclone twice into the pALC14 vector as described in the 'construct' section. The probe was then derived from the pGEMT intermediate vector using SP6 RNA polymerase from the DIG RNA labelling kit (Roche). Hybridisation was carried out at a temperature of 62°C.

## Kinase assay

The identification of a generic in-vitro substrate was performed using a 'cold' kinase assay with 10 μg dephosphorylated MBP, 40 μg histone H1, 1 μg histone cores H2A, H2B, H3, H4, 20 μg dephosphorylated casein, 1 μg β-Casein, or 1 μg of the recombinant *Mus musculus* Caspase 9 (Casp9, first 200 amino-acids - Cloud-Clone Corp, #RPA627Mu01) as substrates in kinase buffer at pH 7.5 (50 mM of MOPS pH 7.5, 100 mM NaCl, 10 mM MgCl2, 10 mM MnCl2) in 20 μl final volume and in the presence of 250 μM of adenosine- triphosphate (ATP)- γ -S (*Allen et al., 2007*). After 30 min incubation at 37°C, the phosphotransferase reaction was stopped by an incubation for 10 min at 95°C, immediately followed by 2 hr of incubation in presence 5 mM PNBM at 20°C to initiate the alkylation reaction as described in *Allen et al. (2007)*. The reaction was then stopped by adding Laemmli loading buffer. Reaction mixtures were separated by SDS–PAGE and transferred to PVDF membrane. Protein loading was revealed either by ponceau staining (substrate) or western blotting (kinase) using the αBB2 antibody. The phosphotransferase activity was revealed by western blotting using the αThioP antibody, as previously described (*Allen et al., 2007*).

The structure/function analysis was performed in a 'hot' kinase assay, using 5 percent of the TY-YFP-DYRK purified protein, incubated on a shaker for 25 min at 37°C with 0.5 μg of the Casp9 as

substrate, 200 mM of ATP, 50 mM of MOPS pH 7.5, 100 mM NaCl, 10 mM MgCl2 and 1 mCi [γ−32P]-ATP (3000 Ci/mmol) in a final volume of 20 µl. The phosphotransferase reaction was then stopped by adding Laemmli loading buffer and boiling at 95°C for 5 min. Reaction mixtures were separated by SDS–PAGE and transferred to PVDF membrane. Protein loading was revealed either by ponceau staining (substrate) or western blotting (kinase) using the αBB2 antibody. $^{32}$P incorporation was monitored by exposing the membrane on an X-ray sensitive film (Roche) at −80°C. After exposure, the bands corresponding to Casp9 were excised from the PVDF membrane and Cherenkov radiation was quantified by a scintillation counter using the $^{32}$P program.

Kinase assays to identify substrates of the DYRK kinase was performed as follows: 15 percent of the TY-YFP-DYRK purified protein, incubated on a shaker for 30 min at 37°C with 1 mg of parasite lysate as substrate (see the phosphoproteomic section for the details), 200 mM of ATP, 50 mM of MOPS pH 7.5, 100 mM NaCl, 10 mM MgCl2. The kinase reaction was stopped by boiling samples for 5 min at 95°C.

## RT-qPCR

RNA preparation was performed using the Qiagen RNA extraction kit, according to manufacturer's instructions. 1 µg of RNA was treated with RQ1 RNase-free DNase (Promega) for 2 hr at 37°C before heat-inactivation. cDNA synthesis was performed using the SuperScript III Reverse Transcriptase (Invitrogen), according to manufacturer's instructions, in presence of oligo(dT)$_{20}$ (Invitrogen) and 500 ng of RNA. Real time PCR was performed using a LigthCycler 96 (Roche). Oligonucleotides MC057/058 and MC059/060 amplified ~120–150 bp fragments of TbDYRK or the YFP tag, respectively. Oligonucleotides MC055/056 (Ma et al., 2010), recognizing a fragment of GPI8, were used as an endogenous control for normalisation. PCRs were set up in triplicate, with each reaction containing 10 µL of Luna Universal qPCR Master Mix (New England BioLabs), 300 nM of each oligonucleotides, 5 µL of cDNA (diluted 1/10) in a final volume of 20 µL. PCR conditions were as follows: 1 cycle of 50°C for 2 min, 1 cycle of 95°C for 10 min, followed by 50 cycles of 95°C for 15 s and 58°C for 1 min. Final melting curve was obtained by gradient increase temperature from 65°C to 95°C.

Ct values were normalised with the internal loading control GPI8. To allow relative quantifications, results were then compared to the mean value obtained for the DYRK gene in the WT strain, that has been set at 100% of expression.

## Cell viability assay

$1 \times 10^5$ parasites per mL were incubated in HMI-9 at 37°C in 5% CO2, for 48 hr, in the presence of 2 µg/mL of doxycycline and in the presence or absence of 100 µM of the cell permeable cyclic AMP analogue, 8-pCPT-cAMP. After 48 hr, 100 µL was transferred into white bottomed 96-well plates. 10 µl alamarBlue (Bio-Rad) was added to each well and the plate was incubated for 4 hr in the same conditions. Fluorescence was read using a FLUOstar OPTIMA fluorimeter (BMG Labtech) at 544 nM excitation/590 nM emission. The mean of fluorescence values were normalised relative to the untreated control (-cAMP) for each strains.

## Phosphoproteomic analysis

The proteomic analysis comparing the phosphoproteome of the two strains *T. brucei* EATRO 1125 AnTat 1.1 90:13 WT or DYRK KO was performed exactly as described in McDonald et al. (2018). Briefly, samples were extracted from two replicates for each cell line. For each, $2.7 \times 10^8$ cells were lysed (4% SDS; 25 mM Tris(2-carboxyethyl)phosphine (TCEP) (Thermo); 50 mM N-ethylmaleimide (Thermo); 150 mM NaCl;1x PhosSTOP phosphatase inhibitor (Roche); 10 mM Na$_2$HPO$_4$ pH6) and then chloroform:methanol precipitated. Tryptic digestion was performed at 37°C with 8 µg trypsin (Pierce). Labelling and mass spectrometry were carried out at the FingerPrints Proteomics Facility at the University of Dundee. The samples were purified by solid phase extraction and then quantified by bicinchoninic acid assay prior to labeling with isobaric tandem mass tags (6-plex TMT). The samples were pooled and divided into fractions using hydrophilic interaction liquid chromatography. Statistical analysis of the fold change in phosphorylation between DYRK KO and *T. brucei* AnTat replicates was performed using the R package limma (Phipson et al., 2016; Ritchie et al., 2015).

The substrate identification analysis was performed with $2 \times 10^9$ of procyclic cells. Lysates were generated using 4% SDS buffer (4% SDS, 25 mM Tris, 50 mM N-Ethylmaleimide (NEM), 150 mM

NaCl, 10 mM Na2HPO4, Protease inhibitor cocktail Roche mini EDTA-free, 25 U/mL Benzonase, pH6), incubated 30 min on ice, sonicated and heat inactivated for 5 min at 95℃. Next, samples were treated with Lambda Phosphatase (~1 U/μg of proteins) for 1 hr at 30℃, followed by another heat inactivation step of 5 min at 95℃. The kinase assay was then performed in three replicates with the active kinase NM or the inactive one (H866A), purified from insect cells, as described in the kinase assay section. To remove any trace of SDS, samples were precipitated with ice cold acetone, with a ratio of acetone/sample of 90%/10% and incubated at −20 overnight. The next day, samples were pelleted by centrifugation for 10 min at 4℃, the supernatant discarded, and washed again with 1 ml of ice-cold acetone and re-pelleted at four degrees. The following steps were performed at the proteomic facility of the University of Edinburgh. Dried pellets were resuspended in 8M urea and a protein assay performed (Bradford Biorad). One milligram of protein extract was digested (enough for two phosphopeptide enrichments). Protein denaturation and reduction was performed in 2M urea, 25 mM ammonium bicarbonate and 5 mM dithiothreitol (DTT). Samples were kept at room temperature for 30 min before cysteine alkylation in 12.5 mM iodoacetamide for 1 hr. Ten micrograms of trypsin were added and digestions were performed overnight at room temperature. Peptide extracts were then cleaned on an SPE reverse phase Bond Elut LMS cartridge, 25 mg (Agilent). The samples were split into five hundred microgram aliquots and dried under low pressure (Thermo Jouan Speedvac) and stored at −20℃.

Samples were then resuspended with 25 μl of 0.5M lactic acid/50% ACN and sonicated, prior phosphopeptide enrichment by adding 8 μl of resin (100 μg/μl of 10 μm TiO beads in isopropanol - dried down). Following this, samples are mixed and incubated overnight at room temperature, with shaking. The sample/TiO2 mix was then transferred to a small filter and spun for 5 min. 25 μl of 0.5M lactic acid/50% ACN was added to the spin tip and then spun for 1 min at 5000 rpm in a microfuge. The resin washed with first 25 μl 0.5M lactic acid/50% CAN and then with 200 μl 80% ACN/0.1% TFA and spun for 1 min at 5000 rpm in a microfuge. Three additional washes were performed with 200 μl 50% ACN/0.1% TFA and spin for 1 min at 5000 rpm, followed by two washes with 200 μl 80% ACN/0.1% TFA and spin for 1 min at 5000 rpm. The first elution was performed as follows – 2 times with 50 μl 50 mM KH2PO4; the 2nd elution was as follows - 50 μl 2M Ammonia; and the third elution comprised - 50 μl 80% ACN/0.1% TFA. The samples were dried under low pressure and reconstituted in 100 μl Buffer A (2%Acetonitrile in water 0.1%formic acid) before clean-up. Clean-up of the eluted peptides was carried out with C18 membrane tips and then dried under low pressure. Once dried, the samples were reconstituted in 7 μl 2% Formic Acid and filtered using a 0.45 μm filter in preparation for MS.

Nano-ESI-HPLC-MS/MS analysis was performed using an on-line system consisting of a nano-pump (Dionex Ultimate 3000, Thermo-Fisher, UK) coupled to a QExactive instrument (Thermo-Fisher, UK) with a pre-column of 300 μm x 5 mm (Acclaim Pepmap, 5 μm particle size) connected to a column of 75 μm x 50 cm (Acclaim Pepmap, 3 μm particle size). Samples were analysed on a 90 min gradient in data dependent analysis (one survey scan at 70 k resolution followed by the top 10 MS/MS). The gradient between solvent A (2%Acetonitrile in water 0.1%formic acid) and solvent B (80% acetonitrile-20% water and 0.1% formic acid) was as follows: 7 min with buffer A, over 1 min increase to 4% buffer B, 57 min increase to 25% buffer B, over 4 min increase to 35%, over 1 min increase to 98% buffer B and continuation under those conditions for 9 min, switch to 2% buffer B over 1 min and the column was conditioned for 10 min under these final conditions. MS/MS Fragmentation was performed under Nitrogen gas using high energy collision dissociation in the HCD cell. Data was acquired using Xcalibur ver 3.1.66.10.

Data from MS/MS spectra was searched using MASCOT Versions 2.4 (Matrix Science Ltd, UK) against a *Trypanosoma brucei* database with maximum missed-cut value set to 2. The following features were used in all searches: i) variable methionine oxidation, and S/T/Y phosphorylation, ii) fixed cysteine carbamidomethylation, iii) precursor mass tolerance of 10 ppm, iv) MS/MS tolerance of 0.05 Da, v) significance threshold (p) below 0.05 (MudPIT scoring) and vi) final peptide score of 20. Progenesis (version 4 Nonlinear Dynamics, UK) was used for LC-MS label-free quantitation. Only MS/MS peaks with a charge of 2+, 3+ or 4+ were considered for the total number of 'Feature' (signal at one particular retention time and m/z) and only the five most intense spectra per 'Feature' were included.

## Bioinformatics approaches

DYRK orthologues of *Leishmania spp* and *Trypanosoma spp* genes and protein sequences were retrieved from the web database TriTrypDB (http://tritrypdb.org/tritrypdb/) (*Aslett et al., 2010*). Homology searches were carried out using BLAST with the default BLOSUM-62 substitution matrix (*Altschul, 1997*), and pattern recognition analysis using the program PRATT v2.1 (*Jonassen, 1997*). Multiple sequence alignments were performed using the built-in algorithm ClustalXv2. Additional sequence analyses were carried out using the Jalview program (*Waterhouse et al., 2009*). Statistical analysis and data plotting were performed using Rstudio software (http://www. rstudio.org/) and R language (R Development Core Team (2005). R: A language and environment for statistical computing. R Foundation for Statistical Computing, Vienna, Austria. ISBN 3- 900051-07-0, URL: http://www. R-project.org).

## Statistical analyses

For the analysis of phenotypes, three to five animals per treatment were routinely used, with pilot and independent replicates confirming observed responses. With effects sizes similar to those previously observed for RNAi mediated loss of developmental competence (0.637 to 1.804; e.g. *Mony et al., 2014*) a sample size of three to five animals per group (+ or − DOX) , or a total of six to ten, allows 80% power for test genes. Data were examined before analysis to ensure normality and that no transformations were required. Global proteomic data analyses were carried out using limma with a moderated t-test. Phosphoproteomic of the substrates of NM and the H866A mutant kinase in cell lysates was carried out using a standard t-test. $P$ values of less than 0.05 were considered statistically significant. Blinding was not carried out.

## Acknowledgements

This work was funded by a Marie Sklodowska Curie postdoctoral fellowship to MC (proposal number 65470) and a Wellcome Investigator award (103740/Z14/Z) and Royal Society Research merit award (WM140045) to KRM.

## Additional information

### Funding

| Funder | Grant reference number | Author |
| --- | --- | --- |
| Wellcome | 103740/Z14/Z | Keith Matthews |
| Royal Society | WM140045 | Keith Matthews |
| European Commission | 65470 | Mathieu Cayla |

The funders had no role in study design, data collection and interpretation, or the decision to submit the work for publication.

### Author contributions

Mathieu Cayla, Conceptualization, Data curation, Formal analysis, Investigation, Methodology; Lindsay McDonald, Conceptualization, Data curation, Formal analysis, Investigation; Paula MacGregor, Formal analysis, Investigation; Keith Matthews, Conceptualization, Formal analysis, Supervision, Funding acquisition, Project administration

### Author ORCIDs

Mathieu Cayla (ID) http://orcid.org/0000-0002-3731-7947
Paula MacGregor (ID) https://orcid.org/0000-0003-0919-3745
Keith Matthews (ID) https://orcid.org/0000-0003-0309-9184

### Ethics

Animal experimentation: Animal experiments in this work were carried out in accordance with the local ethical approval requirements of the University of Edinburgh and the UK Home Office Animal (Scientific Procedures) Act (1986) under licence number 60/4373.

### Decision letter and Author response

Decision letter https://doi.org/10.7554/eLife.51620.sa1
Author response https://doi.org/10.7554/eLife.51620.sa2

## Additional files

### Supplementary files

• Supplementary file 1. List of primers used in the study.

• Supplementary file 2. Results of the phosphoproteomic analysis comparing DYRK-/-cells to WT AnTat1.1cells. Sheet 1 KOvsWT_|FC| > 1.5_p<0.05 presents the peptides with an absolute value of FC > 1.5 and a p-value<0.05. Sheet 2 GOenrichDownKO indicate the GO term enrichment of biological functions of proteins presenting a down regulation of phosphorylation in the DYRK-/-cell line. Sheet 3 GOenrichUpKO indicate the GO term enrichment of biological functions of proteins presenting an up regulation of phosphorylation in the DYRK-/-cell line. Yellow colour indicates the proteins for which further analysis have been performed.

• Supplementary file 3. Results of the phosphoproteomic analysis the phosphorylation of cell lysates by the active DYRK non-mutated NM to the inactive mutant H866A. Sheet 1 NMvsH866A_| FC| > 1.5_p<0.05 presents the peptides with an absolute value of FC > 1.5 and a p-value<0.05. Sheet 2 GOenrichUpNM indicate the GO term enrichment of biological functions of proteins presenting a phosphorylation in presence of the NM. Yellow colour indicates the proteins for which further analysis have been performed.

• Supplementary file 4. List of proteins (sheet 1) and peptides (sheet 2) common in both analyses with an absolute value of FC > 1.5. Green colour indicates the statically significant protein/peptides (p-value<0.05).

• Supplementary file 5. p-Values. Sheet 1, p-values for the kinase activity assay. Sheet 2, p-values for the RT-qPCR analysis, upper table comparing the expression level of the gene TbDYRK and the lower table the YFP gene. Statistic calculation were performed using a pairwise t-test, considering non-equal variances, and apply the holm correction.

• Transparent reporting form

### Data availability

All data generated or analysed during this study are included in the manuscript and supporting files.

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
