## [Decision Letter]

**Acceptance summary:**

African trypanosomes undergo a complex life cycle in the mammalian host and in the transmitting tsetse fly. Curiously, the parasite development appears to be characterized by successions or proliferative and cell cycle-arrested stages. The best characterized transition between proliferative and arrested forms occurs in the mammalian host, where dividing slender trypanosomes differentiate to stumpy forms. This arrest occurs in response to quorum sensing, most probably in order not to prematurely overwhelm the host by high parasite numbers. In this manuscript the authors detail the potential role of a very unusual DYRK kinase, which could be a key player in the regulation of the differentiation process, and that connects the regulation of gene expression with signal transduction. This is an exciting new aspect not only for trypanosome biology, but also for our understanding of the evolution of dual specificity kinases.

**Decision letter after peer review:**

Thank you for submitting your article "An atypical DYRK kinase connects quorum-sensing with posttranscriptional gene regulation in *Trypanosoma brucei*" for consideration by *eLife*. Your article has been reviewed by three reviewers, and the evaluation has been overseen by a Reviewing Editor and Dominique Soldati-Favre as the Senior Editor. The following individuals involved in review of your submission have agreed to reveal their identity: Álvaro Acosta Serrano (Reviewer #2).

The reviewers have discussed the reviews with one another and the Reviewing Editor has drafted this decision to help you prepare a revised submission.

Summary:

This manuscript characterises the role of an atypical DYRK kinase in *Trypanosoma brucei* proliferative slender to cell cycle-arrested stumpy differentiation. Biochemical analysts of the kinase, together with in vivo and in vitro experiments provide evidence that this enzyme holds the balance between parasite proliferation and differentiation. Key amino acid residues of the divergent DYRK kinase are identified, which are essential for slender to stumpy formation. The authors further identify kinase substrates of the kinase through phosphoprotemics and recombinant protein phosphorylation of parasite lysates. These experiments suggest that the RNA binding ZC3H20 is essential for stumpy differentiation, and phosphorylation of the residue T283 is involved in the pathway.

The reviewers acknowledge the broadness and overall quality of the data, especially the mechanistic insights provided, down to the amino acid. The work will not only be of great interest for parasitologists, but also for the general audience interested in kinase signalling.

Despite the overall positive evaluation, a few critical points were raised that need to be addressed before publication.

Essential revisions:

1) The statistics needs to be improved throughout the manuscript. Standard deviations should be provided for all descriptive data.

2) An apparent discrepancy between the bioassay data and the expression levels of the targets needs to be solved. Specifically, the expression of the NM form seems to accelerate the slender to stumpy transition kinetics, even though the transcript levels are similar to wildtype levels. If the levels of the active kinase are the same in both, how can the difference in the bioassay with NM cells be explained? This same concern questions interpretation of the data with the TbDYKR mutants. The expression of kinase inactive forms leads to effects on growth that are similar to wildtype forms, or following ectopic expression of the NM form. In Figure 3A the growth curve for the inactive S472A is the only one that looks like what might be expected, i.e. that for the KO cell line. These discrepancies need to be explained.

3) How was the relative proportion of DYRK and YFP mRNA calculated? Are these data absolute quantifications or relative values? How was the relative quantification of one transcript by normalisation to another (in addition to an internal control) done?

4) It is unclear how RNA levels of TbDYRK can be post-transcriptionally controlled when an ectopic gene is inducibly expressed. Are the native UTRs present on the gene, or is it suggested that this regulation is occurring by regulatory motifs within the coding sequence?

5) The ZC3H20T283A/- lines are shown to be no longer differentiation competent. However, it is unclear how and in which order these cell lines were generated. Specifically, has the differentiation competent ZC3H20T283A/+ line been generated before knocking out the second allele?

---

## [Author Response]

Summary:This manuscript characterises the role of an atypical DYRK kinase in Trypanosoma brucei proliferative slender to cell cycle-arrested stumpy differentiation. Biochemical analysts of the kinase, together with in vivo and in vitro experiments provide evidence that this enzyme holds the balance between parasite proliferation and differentiation. Key amino acid residues of the divergent DYRK kinase are identified, which are essential for slender to stumpy formation. The authors further identify kinase substrates of the kinase through phosphoprotemics and recombinant protein phosphorylation of parasite lysates. These experiments suggest that the RNA binding ZC3H20 is essential for stumpy differentiation, and phosphorylation of the residue T283 is involved in the pathway.The reviewers acknowledge the broadness and overall quality of the data, especially the mechanistic insights provided, down to the amino acid. The work will not only be of great interest for parasitologists, but also for the general audience interested in kinase signalling.Despite the overall positive evaluation, a few critical points were raised that need to be addressed before publication.

Thank you for the positive evaluation of the manuscript. We have carried out extensive revisions and additional experiments to address the referees’ helpful suggestions.

Essential revisions:1) The statistics needs to be improved throughout the manuscript. Standard deviations should be provided for all descriptive data.

Although error bars (SEM) were provided throughout the original manuscript we have now been through the manuscript to ensure these are included in all cases (mRNA levels, kinase activity). We have also included individual raw datapoints in the histograms for mRNA and kinase activity to aid visualisation. The continuous data represented in the growth curves means that we have not provided p values at all points, but error bars are included throughout (as they were in the original manuscript). A Table of p values for each comparison presented in the manuscript is now included as a Supplementary file 5.

2) An apparent discrepancy between the bioassay data and the expression levels of the targets needs to be solved. Specifically, the expression of the NM form seems to accelerate the slender to stumpy transition kinetics, even though the transcript levels are similar to wildtype levels. If the levels of the active kinase are the same in both, how can the difference in the bioassay with NM cells be explained? This same concern questions interpretation of the data with the TbDYKR mutants. The expression of kinase inactive forms leads to effects on growth that are similar to wildtype forms, or following ectopic expression of the NM form. In Figure 3A the growth curve for the inactive S472A is the only one that looks like what might be expected, i.e. that for the KO cell line. These discrepancies need to be explained.

We interpret the enhanced differentiation response of the NM cell line as being due to elevated protein levels of the TbDYRK. The ectopic copy is provided with an aldolase 3’UTR and so its expression can show distinct regulation to the endogenous copy. We have now made this more explicit in the revised manuscript. Unfortunately, however, we cannot detect TbDYRK protein either using antibody or using the epitope tag in any of our cell lines, we believe this being caused by the low levels of overall expression of the protein. Indeed, this is why we reverted to analysing mRNA levels by RT-qPCR as a mechanism to monitor relative expression of the transgenes. With respect to the expression of inactive mutants (or indeed some other mutants that retain activity) the retention of a differentiation response is because, in all cases, the ectopic genes were expressed in the context of the endogenous gene. This was detailed in the original manuscript. Given this, in fact the S472A mutant expression did not conform to expectation. Instead the mutant appears to interfere with the normal function of the endogenous wild type TbDYRK in a dominant negative response that perturbs differentiation, potentially through interference in the interaction of TbDYRK with binding partners. This was detailed in the original manuscript.

3) How was the relative proportion of DYRK and YFP mRNA calculated? Are these data absolute quantifications or relative values? How was the relative quantification of one transcript by normalisation to another (in addition to an internal control) done?

We have added the quantification method details to the revised Materials and methods section.

4) It is unclear how RNA levels of TbDYRK can be post-transcriptionally controlled when an ectopic gene is inducibly expressed. Are the native UTRs present on the gene, or is it suggested that this regulation is occurring by regulatory motifs within the coding sequence?

The mRNA regulation of TbDYRK is indeed intriguing and we discussed this in the manuscript. Apparently, the regulation correlates more with the differentiation response of the parasites rather than being correlated with the kinase activity of TbDYRK. Explicitly, when pCPTcAMP causes cells to arrest, mRNA levels are regulated, whereas if there is not arrest (even if the treated cells show slowed growth with respect to control cells) less regulation is seen, and overall mRNA levels increase. This suggests to us that the regulation of the endogenous levels of TbDYRK mRNA is mediated through other regulators in the pathway (as we discussed in the manuscript). This feedback regulation must be mediated via the 3’UTR of the TbDYRK transcript since the ectopic gene (with an aldolase 3’UTR) is not subject to the same control. We have now amended the manuscript to be explicit on this point. Exploration of the regulatory mechanism is an interesting and important area of study but is beyond the scope of what can be included in our study which already contains a detailed analysis of the activity and substrates of TbDYRK, in addition to its control of a key differentiation regulator by site-specific phosphorylation.

5) The ZC3H20T283A/- lines are shown to be no longer differentiation competent. However, it is unclear how and in which order these cell lines were generated. Specifically, has the differentiation competent ZC3H20T283A/+ line been generated before knocking out the second allele?

This is a valuable point that addresses whether the loss of differentiation competence is directly linked to the phosphorylation potential of ZCH20. Since the ZC3H20 T283A/- was derived from the ZC3H20 T283A/+ line there does remain the possibility of cell line/transfection/selection mediated effects contributing to the loss of differentiation competence. We think this trivial explanation is unlikely since we have analysed independent cell lines, and all showed the same phenotype; this information is now included in the revised manuscript (Figure 6—figure supplement 6). Moreover, in other experiments in our laboratory ZC3H20 RNAi results in an inducible loss of stumpy formation, confirming its role in developmental capacity.

To attempt to address the referee query more fully however, we have carried out extensive further analysis.

Firstly, we performed a ‘rescue’ experiment using the ZC3H20 T283A/- line, via ectopic expression of wild type ZC3H20 (T283). Transfectants grew slowly and finally only a single cell line began to grow well and could be analysed. However, when its phenotype was examined in vivo the line remained virulent (i.e. it did not restore developmental competence) (Figure 6—figure supplement 9). When examined for the expression of the ectopic gene copy, we found leaky expression of the transgene when uninduced and in slender forms- a scenario that selects against developmental competence because expression of wild type ZC3H20 drives a rapid terminal growth arrest as stumpy forms. We believe the failure of most clones to grow through and the selection of this virulent leaky clone is supportive of the add-back ectopic copy restoring density dependent arrest.

Nonetheless, to explore the question yet further, using CRISPR we created a direct add-back replacement (rather than ectopic expression) of the wild type ZC3H20. Endogenous 3’UTR control signals were also used to attempt to maintain physiological levels of expression (5’UTR signals inevitably differ in the constructs). The resulting cell lines partially restored the wild type phenotype, with enhanced cell cycle arrest at high parasitaemia (Figure 6—figure supplement 7C), although PAD1 expression was not detected (Figure 6—figure supplement 7B). The outcome was supportive of functional rescue with respect at least to density dependent growth arrest but the multiple rounds of transfection and selection (4 rounds) needed to generate the lines potentially selects against full restoration of developmental competence, including PAD1 expression. The lineage of cell line generation is now summarised in Figure 6—figure supplement 8.

Overall, these multiple lines of evidence support our model that TbDYRK regulates ZC3H20 likely through the T283 phosphorylation site. However, the experimental challenges of multiple transfection of pleomorphic lines has prevented definitive demonstration of this despite extensive effort. In consequence we have adapted the text of the Results section and Discussion section to reflect this caution in the interpretation of this element of the results.